# FidlTrack: high-fidelity structure-aware single particle tracking resolves intracellular molecular motion in organelles sensing APP processing

Pierre Parutto [1,2] ✉, Yutong Yuan [3], Valentina Davi[1,2], Roger Pons-Lanau [4], Svenja Ebeling[5], Karnika Gupta[1,2], Francesca Bottanelli [5], Maria F. Garcia-Parajo[4,6], Felix Campelo [4,7], Clemens F. Kaminski [3], Joseph E. Chambers [8,10], Jonathon Nixon-Abell [9,10] & Edward Avezov [1,2] ✉

Single Particle Tracking (SPT) is a powerful technique for elucidating the dynamic behaviours of macromolecules within live cells. However, SPT's application to subcellular environments is hampered by the error-proneness of tracking at high particle velocities and densities and the lack of tools to assess trajectory reliability. Here, we introduce FidlTrack, a methodology that benchmarks and improves SPT fidelity. It contains three modules: a parameter optimiser that uses synthetic ground truth SPT data to determine the fidelity-maximising experimental and tracking settings; Structure-aware tracking, that exploits the information provided by organelle structures to constrain particle tracking algorithms; And a tracking quality evaluator that detects, quantifies and removes error-prone ambiguous track segments. Together these tools allow the rational design of SPT experiments, resolving the motion in tight and convoluted organelles, and provide up to 2-fold enrichment in accurate data. We showcase FidlTrack's utility for reliably tracking proteins in the cytosol, mitochondria and endoplasmic reticulum (ER). Further, we demonstrate its efficacy by analysing ER protein dynamics at exit sites, resolving BACE1 amyloidogenic cleavage of the amyloid precursor protein and characterising the spatiotemporal binding dynamics of an ER-targeted intrabody. FidlTrack is provided as a universal open-access platform that can be incorporated into any SPT pipeline.

Single Particle Tracking (SPT) is a powerful technique for studying dynamic molecular behaviours and interactions within live cells[1]. By following the trajectories of individual fluorescently labelled macromolecules such as proteins, lipids or nucleic acids, SPT can reveal their functional status[2]. Further, the motion patterns of SPT report on the cell's biophysical properties (e.g., viscosity and crowdedness). Thus, the development of SPT has heralded a new era in analyses of molecular processes by overcoming the limitations of traditional ensemble measurements[3,4]. By monitoring one molecular displacement at a time, SPT resolves heterogeneities blurred in bulk approaches (e.g., Fluorescence Recovery After Photobleaching - FRAP, bulk photoactivation)[5]. In addition, advances in optics[6], specific labelling chemistry[7,8], and endogenous protein tagging[9] have considerably increased the scope of SPT applications.

However, to date, SPT applications are confined by the target molecules' properties (e.g., cellular location, size, interactions). Most SPT studies concern plasma membrane molecules, imaged in conditions where they exhibit small frame-to-frame displacements (typically $D < 0.5\,\mu m^2/s$[10–12], e.g., neuronal or immune receptors). The challenge of implementing SPT beyond these cases lies in reconstructing trajectories of multiple, fast-moving objects. Notably, SPT of intracellular proteins entail tracking molecules estimated to move 10-100-fold faster than cell surface proteins[13–17]. In these regimes, keeping track of particles is challenging as several indistinguishable objects can be present within the motion range in the next frame. The problem of identifying displaced objects and linking them correctly in successive frames is universally known as the Multiple Object Tracking (MOT) conundrum[18]. In surveillance or animal behaviour tracking the problem is addressed by using object-identifying features such as size, colour and shape[19,20]. Since all spots generated in SPT are indistinguishable, resolving the MOT problem in this context requires a different approach. Further, the featureless nature of SPT spots makes correct trajectories appear indistinguishable from false tracking.

Therefore, to unlock reliable trajectory reconstruction for intra-organelle tracking, here we establish FidlTrack—a methodology to quantify and maximise the fidelity of trajectories reconstructed from broadly used tracking algorithms. Particularly, we introduce the concept of structure-aware tracking which exploits organelle geometries as a novel source of information to critically improve trajectory accuracy. The first step in FidlTrack development was to create a tool to generate large amounts of synthetic ground truth trajectory data. Using this custom-made data generator, we systematically explore how the speed of particle motion, the particle density, and the frame-to-frame linking distance parameters all affect tracking fidelity. This approach identifies fidelity-maximising parameters for a given experimental situation. The SPT data generated under these optimal settings are then processed by a tool that detects and removes linking errors from ambiguously identified track segments, improving data fidelity. Together these tools allow the rational design of SPT experiments, resolving the motion in tight and convoluted organelles, and provide up to 2-fold enrichment in accurate trajectories after ambiguity removal.

This approach pushes the limits of tolerable densities and speeds at which reliable trajectories can be generated. We used FidlTrack to track proteins in several intracellular compartments: the endoplasmic reticulum (ER) lumen and membrane, the mitochondrial inner matrix and outer membrane, and the cytosol of cortical neuronal axons. Furthermore, we demonstrate that FidlTrack affords the sensitivity to detect subtle changes in protein status from its motion. We exploit this to resolve protein dynamics at ER exit sites, visualise amyloidogenic cleavage of the Amyloid Precursor Protein (APP) by BACE1 and quantify the binding of an ER luminal intrabody to APP.

FidlTrack is available as an open access platform allowing users to find fidelity-maximising parameters for any target of interest and evaluate and improve tracking data reliability using structural awareness.

## Results
### Parameterising tracking fidelity
To generate a rigorous single particle trajectory reconstruction pipeline for intracellular targets, we first sought to characterise how tracking errors were impacted by different experimental conditions. We identified five key parameters influencing tracking fidelity. These include: (i) two characteristics of tracked molecules – their speed, and the geometry of the compartment they inhabit; (ii) two tuneable imaging parameters – the framerate and density of visualised particles in each frame; and (iii) a single post-imaging parameter – the linking distance (the maximum distance a particle is allowed to travel between two frames, Fig. 1a). We set out to develop a methodology for

establishing an optimised set of tuneable parameters, maximising SPT fidelity for any given target (see also Methods section "spots localisation precision"). To this end, we created a synthetic ground truth data generator[21,22], based on single-particle stochastic simulations mimicking fluorescence image recordings (see Methods section "Single-Particle Tracking Evaluation", and Supplementary Video 1). Using this generator, we created ground truth trajectories from which we extracted the spots and fed their spatiotemporal coordinates to a tracking algorithm (LAP tracker from Trackmate[23]). We then compared the reconstructed and ground truth trajectories and quantified the fraction of correctly recovered displacements – herein Fidelity Score (Fig. 1b).

Using this procedure, we explored how the natural and tuneable parameters impact tracking by varying them and reading out the Fidelity Score (see methods section "Comparing ground truth and reconstructed trajectories"). To reduce the dimensionality of the problem, we combined the particle's speed (its diffusion coefficient $D$ in $\mu m^2/s$) and the video framerate ($f$ in 1/s) into a single parameter, that we call the characteristic length $l = \sqrt{D/f}$ in $\mu m$ (e.g., at 20 Hz framerate, $l \approx 50$ nm for $D = 0.05\,\mu m^2/s$ and $l \approx 500$ nm for $D = 5\,\mu m^2/s$, Fig. 1c, Fig. S1a). For Brownian motion, $l$ is the only parameter of the frame-to-frame displacement distribution[24] (see Methods section "Brownian simulations in free space").

Starting with freely moving molecules (freespace) and a fixed spot density, we observed that there existed an optimal linking distance maximising fidelity for each tested characteristic length (Fig. 1d). This optimum appeared at the boundary between two opposing effects at varying linking distances: smaller values resulted in the truncation of the intrinsic displacement distribution (Fig. S1b, note the trough left to the red dots in Fig. 1d), while larger values resulted in erroneous linking of unrelated neighbour particles, assuming a large enough field of view. The existence of an optimal distance was maintained for all tested combinations of parameters, but its value increased with characteristic length and decreased with density (Fig. 1e). The Fidelity Score at the optimal distance was decreased by both higher characteristic lengths and densities (Fig. 1f). These results indicated that the optimal linking distance can be determined based on density and characteristic lengths, reducing the dimensionality of the problem to these two parameters.

Therefore, we mapped out the optimal linking distance and the associated Fidelity Score for a comprehensive set of characteristic length and density values (Fig. 1g). These maps showed that tracking at low to medium characteristic lengths (< 300 nm, typical of the range of motion and imaging speeds of plasma or organelle membrane proteins, Fig. 1c) can be done reliably under a wide range of densities. Crucially, at higher characteristic lengths (typical for soluble proteins), reliability quickly deteriorated with increasing densities (note the darker orange, lower Fidelity Score cells toward the upper right corner of the matrix in Fig. 1g, right). Also notable was the fast decrease of the optimal linking distance with density, at higher characteristic lengths (Fig. 1g, left). In such cases, one should expect to miss the larger displacements from the population due to truncation errors. Further, we found that, although increasing spot density can proportionally add meaningful displacements at low characteristic lengths, there was little to no gain from increasing density at higher characteristic lengths (Fig. 1h, see Methods section "Evaluating the density response of an SPT experiment"). Other broadly used tracking algorithms, including nearest / furthest neighbour and u-track[25], yielded similar results (Fig. S1c–e, see Methods section "Tracking from simulations"). Furthermore, more complex modes of motion such as sub- or super-diffusion (fractional Brownian motion with anomalous exponents $\alpha = 0.5$ or 1.5) or mixed diffusional populations (with different proportion of the diffusion coefficients $D_1 = 0.1\,\mu m^2/s$ and $D_2 = 3\,\mu m^2/s$) also possessed optimal linking distances. Moreover, these scenarios obeyed the same trends as Brownian motion – faster particles and increased

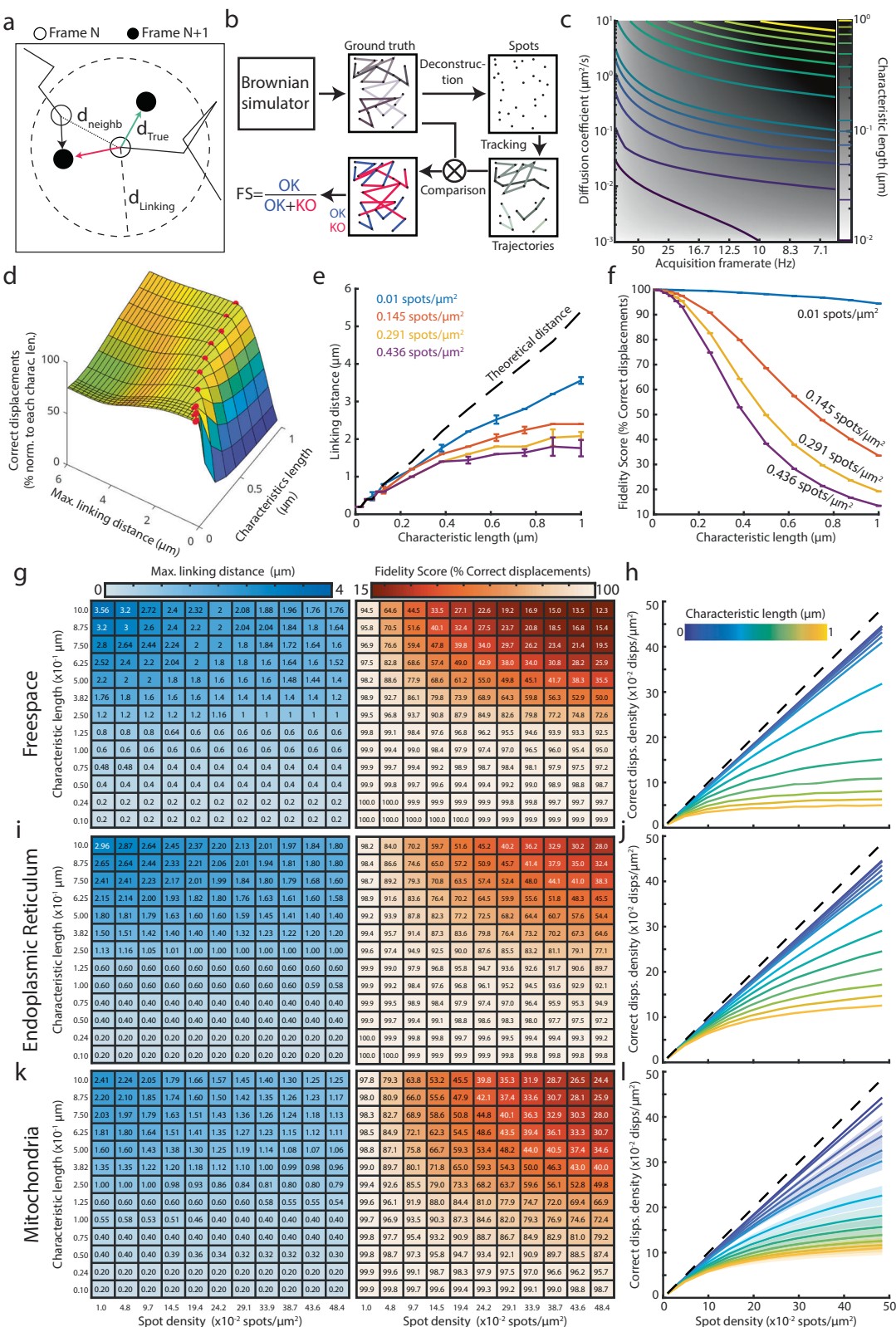

density result in similar loss of tracking fidelity (Fig. S1f–h, see Methods section "Fractional Brownian motion simulations"). Thus, tracking fidelity at high characteristic lengths strongly depends on density, imposing a limit to SPT's capacity to spatially resolve molecular motion.

To explore the impact of organelle structures on the optimal linking distance and fidelity, we applied the same procedure, but using simulations where particle motion was constrained to realistic ER and mitochondria geometries (Fig. S2a–c, also see Methods sections "Simulation in organelles geometries", and Supplementary Videos 2-3). The linking distances and Fidelity Scores for these geometries showed trends comparable to the observations in freespace. However, both geometries showed improved fidelity values and lower sensitivity to density at higher characteristic lengths (Fig. 1i-l). Notably, the Fidelity

**Fig. 1 | Parametrising tracking fidelity. a** Sketch depicting how frame-to-frame object linking is affected by the maximum linking distance for tracking ($d_{Linking}$), the distance to the nearest neighbour at the current frame ($d_{neighb}$), and the intrinsic displacement length of the particle ($d_{True}$). **b** Sketch of our simulation-based SPT fidelity evaluation pipeline. Ground truth trajectories generated by our simulator are deconstructed to extract particle location in each frame and fed to a tracking algorithm (LAP tracker in Trackmate) to reconstruct trajectories, mimicking tracking from experimental recordings. These reconstructed trajectories are then compared to the ground truth to compute the Fidelity Score (see Methods section "Comparing ground truth and reconstructed trajectories"). **c** Characteristic length of the displacement (Rayleigh) distribution for a 2D Brownian motion as a function of the particle speed (diffusion coefficient) and the acquisition framerate (see Methods section "Principles of tracking algorithms"). **d** Percentage of correctly recovered trajectory displacements, normalised to the maximum at the corresponding characteristic length, for a density of 0.291 spots/$\mu m^2$, variable particle speeds and maximum linking distances. The red dots indicate the position of the maximum value at each characteristic length. **e** Evolution (mean ± SEM) of the optimal linking distance as a function of the characteristic length for different spot densities. The black dashed "theoretical distance" line represents the expected distance to capture 99.9% of the displacements for the corresponding 2D Brownian motion (see Methods section "Tracking difficulty increases with characteristic lengths"). **f** Evolution (mean ± SEM) of the optimal Fidelity Score as a function of the characteristic length for different spot densities. The yellow line (density of 0.291 spots/$\mu m^2$) corresponds to the red dots in (**d**). **g, i, k** Maps of the optimal linking distances (left) and associated Fidelity Scores (right) for a wide range of spots densities and characteristic lengths for simulations in freespace (**g**), ER (**i**) or mitochondria (**k**) geometries. **h, j, l** Density (mean ± SEM) of correctly recovered displacements as a function of the spot density for different characteristic lengths (colours) for simulations in freespace (**h**), ER (**j**) or mitochondria (**l**) geometries. Simulations results are averages over 5 independent repeats in freespace and over 5 independent repeats for each of the 6 ER and 5 mitochondria geometries. Source data are provided as a Source Data file.

Score for mitochondria SPT was more sensitive to density at lower characteristic lengths compared to freespace (Fig. 1k). Overall, the presence of geometrical constraints improves tracking fidelity, but the specific shape and amplitude of this improvement depend on the structure's characteristics (Fig. S2d, e).

To enable users to easily determine fidelity-maximising SPT settings (image acquisition framerate, spot density, and linking distance), we incorporated the presented lookup tables into our FidlTrack platform. Prior information on the mobility of the target molecule can be used to further optimise the selection of appropriate settings.

## Ambiguity score-based filtering of low-fidelity displacements

To improve SPT fidelity we sought a strategy to identify error-prone linkages in trajectory reconstructions. In most tested conditions, we noticed that over half of linking errors in FidlTrack simulations arose from ambiguous displacements (Fig. 2a, note that the non-monotonous relationship is caused by a larger fraction of edge errors appearing at higher characteristic lengths and densities). Ambiguous displacements occur when a spot can be linked to more than one spot in the next frame, requiring the tracking algorithm to 'make a choice' (Fig. 2b, see Methods section "Ambiguity scoring")[26]. Although the percentage of ambiguous displacements, which we term the Ambiguity Score, exhibited a large range of values, from 0 to 99% (Fig. 2c and Fig. S3a), it is a good approximation to the Fidelity Score (Fig. S3b) and can be measured directly on acquired data. Removing ambiguous displacements post-tracking (Fig. 2d) can increase the overall fidelity (Fig. 2e) but sacrifices a significant portion of trajectory segments (Fig. S3c), shortening trajectories (Fig. S3d). However, ambiguity removal substantially improved diffusion coefficient estimation (Fig. S3e). This loss of meaningful data can only be marginally compensated by increasing spot densities (Fig. 2f, Fig. S3f, note the appearance of a plateau at lower characteristic lengths than in Fig. 1h, j, l). It can however be compensated by longer acquisitions.

These principles were exemplified by tracking an ER luminal marker (HaloTag[ER])[27] at two different spot density levels in the same COS-7 cell (Fig. 2g, Fig. S3g, Supplementary Videos 4–5). Mirroring the simulations, in these data, the Ambiguity Score was correlated with spot density (Fig. 2h) and more trajectories were affected by ambiguity removal in the denser dataset (Fig. 2i). Further, the anisotropic distribution of ambiguous displacements was apparent, with higher ambiguities appearing in the perinuclear ER (Fig. 2j), likely due to local higher spot density. Upon ambiguity removal, around half the displacements were lost from the higher-density recording compared to 10% in the lower-density regime (Fig. 2k).

Thus, the ambiguity scoring module of FidlTrack serves as a benchmark for SPT imaging data fidelity, identifying and removing error-prone trajectory segments. Particularly, ambiguity maps (Fig. 2j) allow to detect and selectively remove data from high-ambiguity regions (e.g., perinuclear region in Fig. 2j). Unlike changing global tracking parameters, removal of ambiguous trajectories is local, cleaning up dense regions without compromising fidelity in sparser areas.

## Structure-aware tracking improves SPT fidelity and resolution

We hypothesised that a substantial increase in the accuracy of organellar SPT can be attained by accounting for their specific structures. This information is readily available by imaging an organelle marker either pre-/post-SPT acquisition or simultaneously in two channels using spectrally separated fluorophores. As organelle geometries such as tubulated ER or mitochondria limit the range of possible particle motion, including this information in tracking is expected to improve fidelity and reduce ambiguity of frame-to-frame linking.

To introduce structural awareness, we first recovered the binary masks of organelles' shapes by AI-aided segmentation[28] of fluorescence images of their markers (e.g., mitochondria matrix or ER-targeted mEmerald, Fig. 3a). We used the resulting images to build a graph connecting neighbour mask pixels. From this graph, we computed the 'graph distance' $d_G$ between all connected pairs of points - the distance along their shortest path, approximating their distance along the structure (Fig. S4a). Crucially, the graph distance is infinite between disconnected structures and always greater or equal to the (freespace) Euclidean distance $d_E$ (see Methods section "Structure-aware Tracking"). This provided the basis to identify and remove impossible trajectory segments – connections between proximal non-contiguous structures (red segments in Fig. 3b, c).

In a scenario designed to explore the maximal benefits of structure-aware tracking, we simulated particles in an environment of close-by disconnected tubules (Fig. 3c, mimicking neurites). In this case, structure-aware tracking achieves up to 50% gain in fidelity (Fig. 3d) and 77% reduction of ambiguity (Fig. 3e). We showcased such geometry by tracking a cytoplasmic HaloTag probe in neurites of human iPSC-derived cortical neurons[29] (Fig. 3f, Fig. S4b, Supplementary Video 6). While conventional tracking was riddled with erroneous trajectory segments (Fig. 3g, crossing prohibited zones between separate neurites). Structure-aware tracking prevented these errors (Fig. 3h). This allowed the resolution of proximal individual neurites and substantially reduced tracking ambiguities (Fig. 3i).

For simulations in more densely interconnected geometries such as that of compact ER and mitochondria, the gains were milder, up to 5% improvement in Fidelity (Fig. 3j, Fig. S4c) and 50% reduction in ambiguity (Fig. 3k). Notably, the highest fidelity gains in mitochondria are apparent at moderate particle densities indicating that at higher densities most errors are not rectifiable through accounting for structure. Despite the raw gains in fidelity being mild in these cases, the

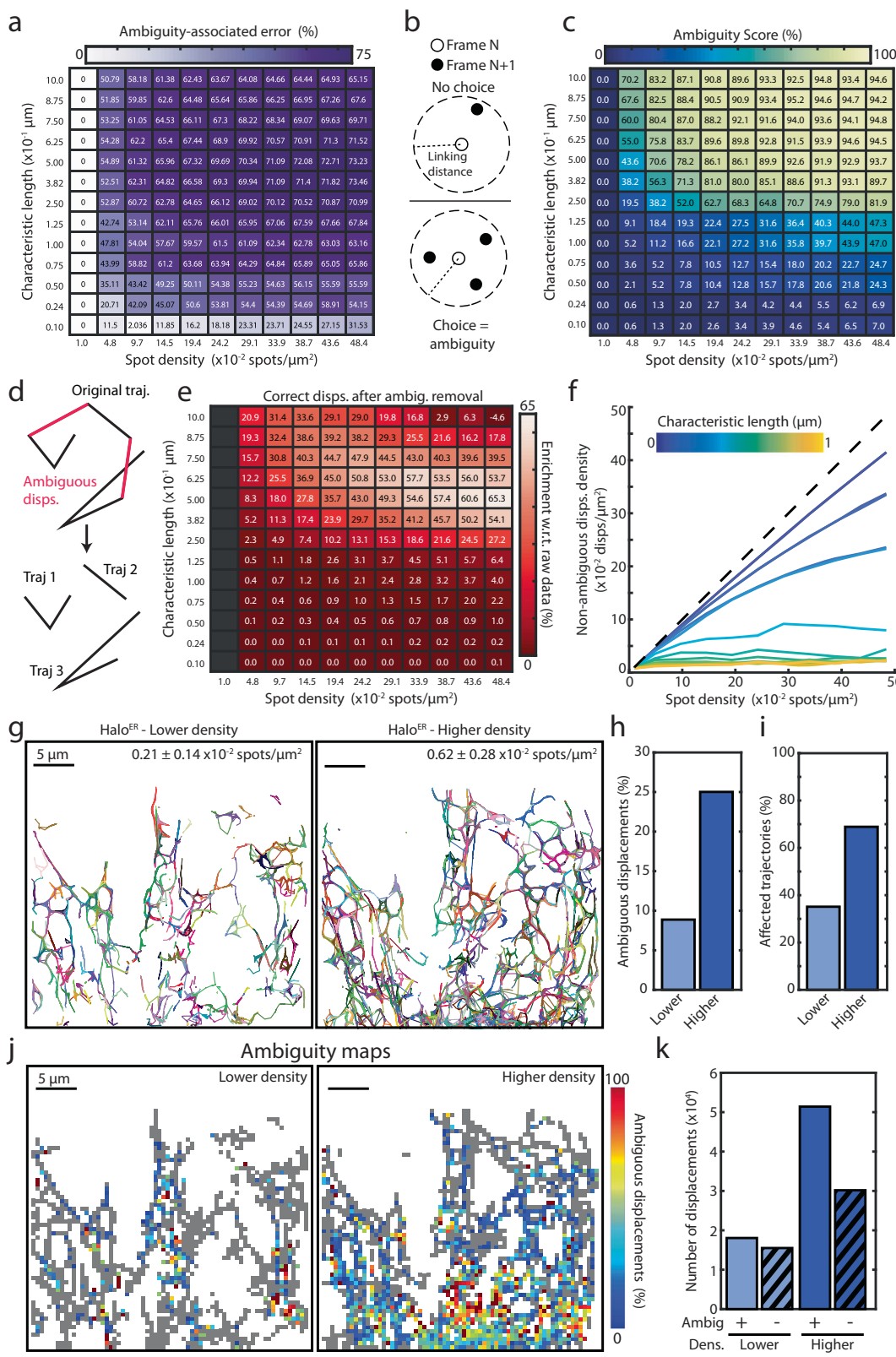

large reduction in ambiguity leads to a substantial increase of the amount of retained non-ambiguous data. Consequently, structure-aware tracking allows to recover more non-ambiguous displacements: up to 3.7-fold in ER and up to 24.6-fold in mitochondrial geometries for a given spot density (Fig. 3l, Fig. S4d). These results show that in different tracking scenario the three main modules of FidlTrack – parameter optimisation, ambiguity-removal and structure-awareness – contribute to different extents to improving fidelity.

Thus, by exploiting the structural constraints imposed by organelles on particles' motion, structure-aware tracking improves SPT fidelity and reduces ambiguity, increasing the capacity to resolve molecular motion patterns in space-time.

**Fig. 2 | Ambiguity scoring for evaluating and improving tracking fidelity.**
**a** Percentage of linking errors occurring at ambiguous displacements in freespace simulations at the optimal linking distance for a wide range of characteristic lengths and spot densities. **b** Sketch depicting the ambiguity concept for a displacement: non-unambiguous displacements occur when only one spot is present in the next frame within the linking distance of the starting spot, leaving a single option for the tracking algorithm (top); When more than one spots are present in the next frame within the linking distance of the starting spot, the displacement is ambiguous with multiple linking options (bottom). **c** Map showing the Ambiguity Score (percentage of ambiguous displacements) in freespace simulations at the optimal linking distance for a wide range of characteristic lengths and spot densities. **d** Sketch presenting the trajectories splitting procedure for removing ambiguous displacements: each time an ambiguous displacement appears in a trajectory two new trajectories are formed containing all the displacements before and after the ambiguous displacement. **e** Percentage enrichment of correct displacements after ambiguity removal in freespace simulations at the optimal linking distance for a wide range of characteristic lengths and spot densities. **f** Non-ambiguous displacements density (mean ± SEM) as a function of the spots densities for different characteristic lengths (coloured) in freespace simulations. **g** Experimental single-particle trajectories data of an ER luminal probe Halo$^{ER}$ stained with PA-JF646 dye from two successive recordings of the same cell at a "lower" (left) or "higher" (right) spots densities, colour-coded by individual trajectories. Spots appearing outside the structures (Fig. S3g) were removed. **h** Bar plot presenting the percentage of ambiguous displacements for the two recordings from (**g**). **i** Bar plot presenting the percentage of trajectories containing at least one ambiguous displacement for the two recordings from (**g**). **j** Spatial map presenting the percentage of erroneous displacements in each square bin (of size 50 nm) for the "low" (left) and "high" (right) density recordings from (**g**). **k** Bar plot presenting the recovered number of displacements for the two recordings from (**g**) before and after ambiguity removal. The results derived from simulations are averages over 5 repeats in freespace. Source data are provided as a Source Data file.

## Structure-aware tracking in mobile organelles

Next, we applied the structure-aware FidlTrack approach to visualising and analysing molecular motion in the lumen or on the outer membrane of mitochondria and the ER. To this end, we used a dual colour imaging modality whereby widefield imaging of the organelle structure is acquired in the green emission channel, simultaneously with the single-molecule images in the far-red channel (Fig. 4a, Supplementary Videos 7–9). This approach is necessary since the organelles restructure and move during the image acquisition period (Fig. 4b, c, note the differences in organelles' structure between the start/end frames and in the stability maps). Therefore, we introduced an extension to structure-aware tracking that dynamically processes the mask's morphological information across multiple frames (see Methods section "Structure-aware Tracking in dynamic structures"). This recovered up to 40% more spots otherwise lost when using static organelle structure images recorded pre- or post-SPT acquisition (Fig. 4d). Applying the FidlTrack principle in structure-aware mode to mitochondrial tracking revealed distinct trajectory dynamics for a matrix (Mito-targeted HaloTag) and outer membrane (TOMM20-HaloTag) marker (Fig. 4e). Based on these data, we resolved variations in the diffusivities between the organelle compartments (Fig. 4f, Table 1, see Methods section "Fitting of displacement lengths distributions"). Furthermore, structure-aware FidlTrack showed sensitivity sufficient to detect the predicted increase in mobility of an ER membrane marker (SEC61B-HaloTag, Fig. 4g) following ER membrane fluidisation upon treatment of cells with the monounsaturated fatty acid oleate[30] (Supplementary Video 10). Such differences fell within the measurement variability in ensemble FRAP experiments (Fig. S5a, b). Improving SPT fidelity by applying structure-awareness and ambiguity removal increased the resolution of the difference between the two conditions (Fig. 4h) while only marginally impacting the recovered dynamic characteristics of molecules (Table 1).

Provided that organelles can sometimes fluctuate or move rapidly, we incorporated a feature in FidlTrack accounting for the mobility of the container structure. This mode of structure-aware tracking decouples the particle from the organelle motion based on the dynamic structure mask. This is exemplified for a TOMM20 trajectory at the surface of a mobile mitochondrion (Fig. 4i, Supplementary Video 11, see Methods section "Decoupling particle and organelle motion"). These results showcase the capacity of FidlTrack-based structure-aware tracking to improve the quantitative resolution of molecular motion patterns in cellular compartments.

## ER luminal particle behaviour at ER Exit Sites (ERES)

To explore the limits of detection and analysis of rare events using SPT, we evaluated the performance of FidlTrack in detecting a key transient intraorganellar event—the engagement of early secretory pathway proteins with ER exit sites (ERES)[31–33]. Understanding the precise mechanisms governing the sorting or exclusion of proteins during progression through the secretory pathway has been limited by the challenges of visualising individual events of protein entry into ERES. While this problem is in principle amenable to SPT, it requires high-fidelity tracking in a high-density regime to reliably capture these transient events. Therefore, we set out to test how the increased resolution of FidlTrack can aid visualising ERES function. We simultaneously recorded the organelle structure using SEC61B::GFP, the location of endogenous ERES via SEC13 Snap-tagged at its endogenous locus (see Methods section "Lentiviral particles and stable cell-lines production"), along with the single molecule motion of an inert ER luminal protein (Halo$^{ER}$) in live Hela cells (Fig. 5a, Supplementary Video 12). Reconstructing Halo$^{ER}$ trajectories using FidlTrack revealed a clearer ER pattern and lower ambiguity scores compared to generic tracking (Fig. 5b, c). Crucially, without FidlTrack, trajectories at ERES tended to incorporate unrelated localisations, thereby erroneously inflating the number of interacting particles (Fig. 5d–f). FidlTrack reconstruction identified clear differences in retention time for trajectories at ERES compared to those at randomly selected sites outside ERES (Fig. 5g, Fig. S6a,b), with a duration following a double exponential distribution (parameters: $\tau_1 = 10$ ms and $\tau_2 = 56$ ms, Fig. 5h). Interestingly, FidlTrack revealed different behaviours of trajectories at ERES: "flyby" through ERES unimpeded (Fig. 5i), "visiting" ERES for short periods (~200 ms, Fig. 5j), or "dwelling" for longer durations (>2 s, Fig. 5k). Notably, we observed that ERES themselves exhibited different characteristics: a large proportion of ERES were enriched in flyby trajectories (Fig. 5l), while fewer were "sticky", attracting more visiting, longer-dwelling trajectories than flyby trajectories (Fig. 5m, n). Thus, FidlTrack's enhanced accuracy and sensitivity helped resolving the local molecular motion at ERES, exposing otherwise hidden functional features of the cell.

## Visualising amyloidogenic APP cleavage in live cells

To further explore FidlTrack's performance in developing diverse single-particle analysis pipelines, we applied it to study the intracellular cleavage of the Amyloid Precursor Protein (APP). Amyloidogenic processing of APP leads to the accumulation of the aggregation-prone Aβ peptides—a key pathogen in Alzheimer's[34–36]. Visualising this process with subcellular resolution is a challenge addressable by SPT: the transition of cleaved APP fragments to a faster motion-pattern after its liberation from the membrane can be leveraged to distinguish between its intact and cleaved states (Fig. 6a). Given the statistical rarity of APP-cleavage events during the lifetime of a tracked particle, detecting individual events in real-time necessitates high particle densities and renders the tracking extremely sensitive to errors. With this in mind, we set out to visualise the effect on APP of a key AD-relevant protease, the beta-secretase 1 (BACE1)[37] as it passes through the ER.

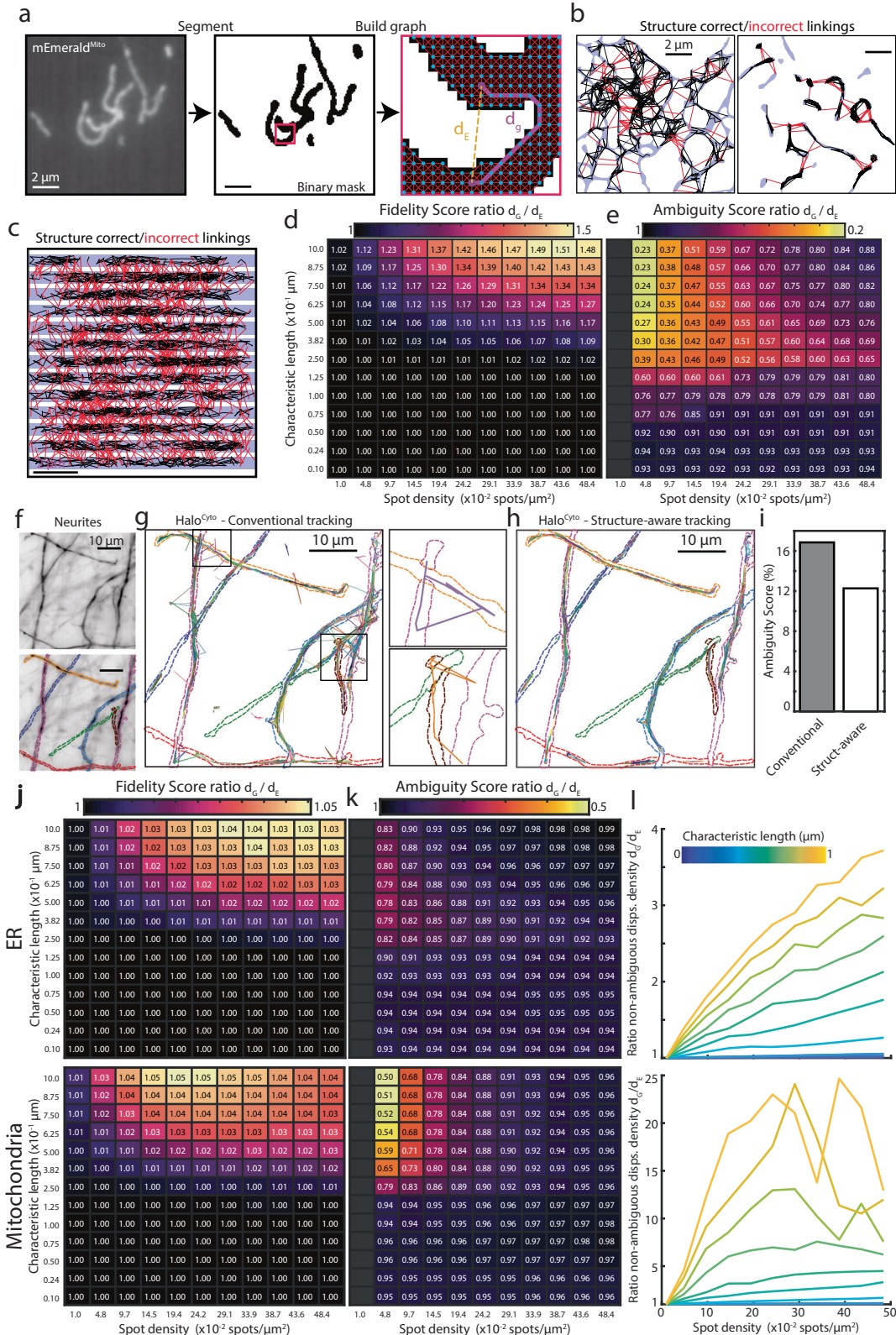

We imaged cells in a multiplexed mode, co-expressing the StayGold ER marker and APP with or without BACE1-GFP over-expression. The APP was retained in the ER by the RUSH system[38] and Halo-tagged at its luminal N-terminus, to allow SPT of its intact and cleaved forms (Fig.6b, Supplementary Videos 13–14). Strikingly, relying on FidlTrack led to a larger difference between BACE1 posi-tive and negative cells than in generic tracking (Fig. 6c, Fig. S7a)

which led to a ~ 2.4-fold underestimation in BACE1 cleavage (Fig. 6d). Further, FidlTrack-based analysis revealed the ultra-rare events of APP trajectories undergoing cleavage in real-time−capturing tran-sitions from slow to fast dynamics within individual trajectories (Fig. 6e, see Methods section "Visualising APP cleavage"). The ana-lysis thus reveals that BACE1 can be active in the early secretory organelle and cleave nascent APP there. This showcases the capacity

**Fig. 3 | Structure-aware tracking improves tracking error and resolution.**
**a** Principle of structure-aware tracking: fluorescence image of the organelle structure (left), segmented image (middle), and the derived graph representing the local pixel connectivity (right) from which the distance between two points along the structure is approximated as the shortest path between their nodes in the graph. Examples of simulated trajectories where structure-incorrect displacements (with graph but not Euclidean distances greater than the linking distance) are displayed in red for an ER (**b**, left), mitochondria (**b**, right) or parallel tubule (**c**) geometries. Ratios of Fidelity (**d**) or Ambiguity (**e**) scores obtained with structure-aware tracking over conventional tracking at their respective optimal linking distances for a wide range of characteristic lengths and spot densities for the parallel-tubule simulations. **f** Total Internal Reflection Fluorescence (TIRF) image of entangled neurites of neurons derived from human induced pluripotent stem cells expressing a cytosolic HaloTag stained with PA-JF646 dye. The image was acquired post-SPT recording with high 405 nm laser power to visualise the marker in bulk. The overlay shows the contour of the identified neurites. **g** Single-particle trajectories data obtained with conventional tracking of the cytosolic probe for the field of view presented in (**f**) and overlaid with the contour of individual neurites. The insets show two different incorrect trajectories spanning multiple neurites. **h** The same single-molecule recording as in panel (**g**) this time tracked with structure-aware tracking. **i** Percentage of ambiguous displacements in the conventional and structure-aware tracked datasets. **j** Same as (**d**) but for simulations in ER (top) or mitochondria (bottom) geometries. **k** Same as (**e**) but for simulations in ER (top) or mitochondria (bottom) geometries. **l** Ratios of the densities of recovered non-ambiguous displacements as a function of the simulated spot density when computing ambiguities with the graph distance over computing ambiguities with the Euclidean distance, for different characteristic lengths (colours) for simulations in ER (top) or mitochondria (bottom) geometries. The results derived from simulations are averages over 3 and 5 independent repeats for the parallel-tubule and for each of the 6 ER and 5 mitochondria geometries, respectively. Source data are provided as a Source Data file.

of FidlTrack to extract key insights into ultra-rare events from complex imaging datasets.

## Resolving the binding status of ER-targeted intrabody based on its motion

To assess the versatility of FidlTrack's applications, we sought a showcase requiring high-fidelity tracking such as resolving changes in individual protein behaviour due to transient interactions. As a study case, we tracked an ER lumen-targeted and Halo-tagged intrabody against APP (Fig. 7a, b). The soluble intrabody binds to an Alfatag[39] epitope which we inserted in the ER luminal portion of APP (Fig. 7b). The intrabody showed almost perfect ER localisation, whereas APP exhibited a mixed ER and punctate, endosome-like pattern (Fig. 7a). This is consistent with APP's known secretory trafficking route, which samples both the ER and endosomes[40].

To establish the binding status of individual intrabodies, we relied on the assumption that bound intrabodies assumed the motion characteristics of their less mobile transmembrane APP targets. Using the FidlTrack methodology, we were able to distinguish bound intrabody trajectories as their average displacement kinetics shifted toward the values measured for APP (Fig. 7c, note the enrichment in slower – blue colour-code range intrabody trajectories in response to APP expression, see also Supplementary Video 15–17, Fig. S7b–d). Computing the distribution of average trajectory displacements over multiple experiments demonstrated that the unbound intrabody and APP motion fitted well to single-component Gaussian distributions (Fig. 7d). Conspicuously, the intrabody motion in the presence of APP appeared as a bimodal mixture of the two (Fig. 7d). Analysing individual intrabody SPT recordings from cells with heterogeneous APP and intrabody expression levels, we could consistently fit their average trajectory displacement distributions to two-component Gaussian models (Fig. 7e, see Methods section "Binding status extraction from intrabody trajectories"). In these models, the motion range of each component coherently represented bound and unbound populations (Fig. 7f). This unmixing revealed a 70–90% bound intrabody fraction across the sampled cells (Fig. 7g), reflecting differences in stoichiometry. Furthermore, we used the fitted models to classify individual trajectories into bound or unbound populations (Fig. 7h), unlocking the visualisation of the target protein's dynamics through intrabody tracking (Fig. 7i). These results exemplify the application of high fidelity, structure-aware SPT to resolve in space and time the interaction patterns and subcellular domains of proteins.

## Discussion

The FidlTrack methodology resolves subcellular molecular motion by improving the fidelity and sensitivity of single-particle tracking (SPT). It starts with detailed parametrisation of fidelity-determining factors based on realistic synthetic data generation. This delineates the limits and requirements of accurate SPT for specific experimental needs. Critical parameters such as framerate and labelling density are optimised based on preliminary mobility estimates of target molecules, ensuring the acquisition parameters maximise tracking fidelity. Optimal data are then processed with FidlTrack to precisely adjust the critical post-acquisition parameter – the linking distance tolerance for accurate trajectory tracking. The basic principles behind FidlTrack carry over to different modes motion: pure Brownian, fractional Brownian motion or mixed Brownian populations, as well as different widely used tracking algorithms. Accordingly, FidlTrack provides generalisable rules to guide experimental design, such as in cases where spot density cannot be maintained constant, our results indicate that selecting a larger linking distance is preferable to a smaller one.

As part of its comprehensive approach, FidlTrack also benchmarks tracking fidelity through the Ambiguity Score—a metric that quantifies the likelihood of erroneous frame-to-frame linkages. While ambiguity can enhance fidelity by filtering out dubious data points, this process may lead to the loss of valuable information, representing a trade-off between accuracy and data completeness. However, the introduction of structure-aware tracking within FidlTrack addresses this challenge by using the physical constraints of cellular architectures to improve tracking accuracy. Structure-aware SPT effectively pushes the limits of what can be achieved with high-fidelity tracking, particularly in complex and densely populated cellular environments. Structure-awareness is particularly efficient at reducing ambiguity and thus increasing the amount of recovered reliable trajectories. The fidelity gain of structure-aware tracking depends on the geometry and scale, e.g., close-by disconnected thin structures producing the highest gains in performance (Fig. 3). Technical factors influencing the successful outcome of this approach include the quality of the structure mask. Loose segmentation reduces the effect of structure-aware tracking, while overly stringent automated segmentation may omit parts of the structure discarding valid particles and thus reducing the amount of recovered information.

Beyond endomembrane organelles, structure-aware tracking can be applied to cytoplasmic targets when their environment can be translated into maskable structures such as neurites (Fig. 3f–i), intracellular obstacles (e.g., aggregates) or other detectable features restricting motion. Thus, geometrical features of the cytoplasm that restrict molecular motion can be leveraged for fidelity improvement following the structure awareness procedures described here.

The versatility and sensitivity of FidlTrack are demonstrated through various case studies, such as tracking ER-targeted intrabodies and mitochondrial markers in moving structures. These examples highlight the method's ability to capture subtle changes in molecular dynamics and interactions, showcasing its applicability in diverse biological settings. Thus, FidlTrack enables access to space-time

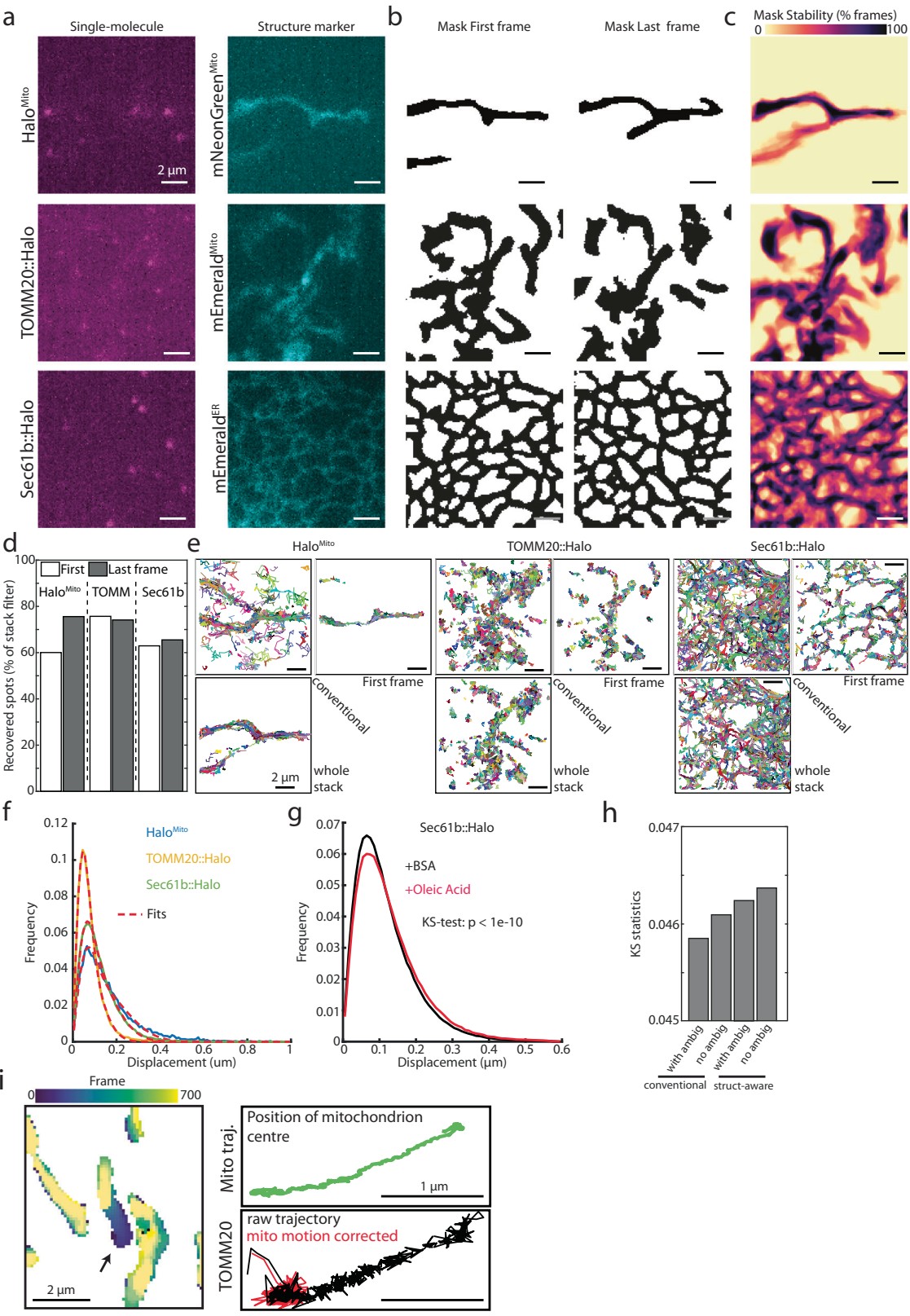

resolved mapping of proteins' functional status, as demonstrated by the analysis of the entry of luminal proteins at ER exit sites, detection of APP cleavage and intrabody biding showcases. However, it is worth noting that such analyses are still limited in resolution, e.g., the detection of intrabody binding requiring a substantial change in mobility upon binding to the target. Uncoupling more subtle mobility changes is an important goal for future work.

The open-source and easy-to-use nature of FidlTrack for experimental parameters optimisation, ambiguity determination and structure-aware tracking, makes this tracking technology broadly accessible. The method can be implemented using widely available commercial microscopes with dual or spectrally split cameras for parallel SPT and organelle structure imaging or single-channel systems recording sequentially. The modular nature of FidlTrack allows its

**Fig. 4 | Improved tracking for detecting subtle dynamical changes in organelles. a** TIRF images of a frame from simultaneously acquired single-molecules (left) and the associated organelle geometries (right) for a mitochondria matrix targeted HaloTag probe and mNeonGreen marker (top), the mitochondria membrane protein TOMM20 and mitochondria matrix targeted mEmerald (middle), and the ER membrane protein SEC61B and ER lumen targeted mEmerald (bottom) in COS-7 cells. **b** Masks extracted via AI-assisted image processing from the first and last frame of the recordings for the mitochondria matrix targeted probe (top), TOMM20 (middle) and SEC61B (bottom) recordings presented in (**a**). **c** Stability maps, representing for each pixel the percentage of total frames in which it is part of a mask, for the mitochondria matrix targeted probe (top), TOMM20 (middle) and SEC61B (bottom) recordings presented in (**a**). **d** Bar plot presenting the fraction of recovered spots relative to the quantity recovered from the entire stack of masks when using the mask of either the first or last frame to perform structure-aware tracking in the different recordings presented in (**a**). **e** Reconstructed trajectories (individually colour-coded) from the datasets presented in (**a**) using conventional tracking (not structure-aware, top, left), structure-aware tracking using the mask from the first frame (top, right) or the whole stack of masks (bottom, left). **f** Plot of the displacement distributions extracted from the trajectories presented in (**e**) fitted to a mixture of two Rayleigh distributions (dashed red lines, see Methods section "Fitting of displacement lengths distributions"). **g** Plot of the pooled displacement distributions for structure-aware ambiguity-removed tracking of SEC61B::Halotag in COS-7 cells after 4 h of BSA or Oleic Acid treatment at 400 μM. Pooling was done over $n = 7/6$ recordings for BSA and oleic acid, respectively, the reported statistics correspond to a two-tailed Kolmogorov–Smirnov test $p < 2.2251e\text{-}308$. **h** Bar plot of the Kolmogorov–Smirnov statistics from (**g**) for either conventional or structure-aware tracking and with or without ambiguity removal. **i** Temporally colour-coded mitochondria positions over 700 frames (left), trajectory of the centre of the pointed mitochondria (right, top) and the raw (black) and mitochondrion motion-corrected trajectory (red) of a TOMM20 receptor moving at the surface of the pointed mitochondrion (right, bottom). Source data are provided as a Source Data file.

application in a wide variety of scenarios and microscopy setups, as exemplified by our showcases. The contribution to fidelity improvement by each module varies depending on the situation.

Thus, FidlTrack benchmarks SPT fidelity and provides means to quantify and improve molecule tracking reliability using all the available information, broadening the range of SPT applications.

## Methods

### Analysis methods

#### Single-particle tracking evaluation

**Brownian simulations in free space.** We consider that molecules follow a Brownian dynamic[41] given by

$$\dot{X}(t) = \sqrt{2D}\dot{\omega}, \tag{1}$$

where $X$ is the 2D position of a molecule at time $t > 0$, $\dot{X}$ is the derivative of $X$ with respect to time, $D$ is the diffusion coefficient (in $\mu m^2/s$) and $\omega$ is a white noise. For such a motion, the distribution of frame-to-frame displacement lengths follows a Rayleigh distribution[24]:

$$f(l) = \frac{l}{2\sigma^2} e^{-\frac{l^2}{4\sigma^2}}, \tag{2}$$

where $\sigma = \sqrt{D\Delta t}$ is the characteristic length in $\mu m$ and $\Delta t$ is the frame acquisition time in s. We simulate Eq. 1 using the Euler-Maruyama method[42] with a discrete simulation time step $\delta t > 0$ as

$$X(t + \delta t) = X(t) + \sqrt{2D\delta t}\eta, \tag{3}$$

where $\eta$ is a 2D vector of independent normally distributed variables of mean 0 and variance 1. To obtain the position of the particles at every frame of the video acquired with an acquisition time $\Delta t \gg \delta t$, the trajectories are subsampled keeping one every $\Delta t/\delta t$ point.

To simulate the mixture of two Brownian motions presented in Fig. S1h, we simulated datasets where trajectories are randomly assigned, depending on the mixture probability, one of the two selected diffusion coefficients. Simulations were implemented in C++ and compiled with g++ (v. 14.2.0).

**Principles of tracking algorithms.** Given an ensemble of spots characterised only by their spatiotemporal coordinates, tracking algorithms assign an identity to each spot based on the frame-to-frame propagation of spot identity. To do this, most tracking algorithms[43] rely on maximising the frame-to-frame transition probability of two spots being linked. This probability for a two-dimensional Brownian particle located at position $X$ at time $t$ to be located at $X'$ at time $t' > t$, is

given by

$$p(\Delta X|\delta t) = \frac{1}{4\pi D\delta t} \exp\left(-\frac{||\Delta X||^2}{4D\delta t}\right),$$

where $||.||$ is the Euclidean norm, $\Delta X = X' - X$, $\delta t = t' - t$ and $D$ is the diffusion coefficient. This equation is maximised for $||\Delta X||^2 \to 0$, thus given multiple particles appearing with the same time delay from a starting particle, the most probable successor is the closest one. Extending this equation to evaluate the identity assignment of $N$ identical and non-interacting particles lead to

$$p(\Delta X_i, i = 1..N|\delta t) = \left(\frac{1}{4\pi D\delta t}\right)^N \exp\left(-\frac{1}{4D\delta t}\sum_{i=1}^{N}||\Delta X_i||^2\right), \tag{4}$$

where $\Delta X_i = X_i' - X_i$. Following Eq. 4, the most probable set of assignments is the one minimising the cumulated sum of displacement lengths $\sum_{i=1}^{N}||\Delta X_i||^2$. The main difference between tracking algorithms lies in the way they minimise this equation. Nearest-neighbour algorithms perform local optimisation, choosing the nearest successor to each spot. This ensures the most probable choice for every assignment but does not necessarily lead to the set of assignments globally minimising Eq. 4, as spots can only be assigned once (typical of greedy optimisation algorithms[44]). On the other hand, tracking algorithms derived from the so called Jaqaman algorithm[25] perform a global minimisation, over all possible assignments, by using the framework of linear assignment problems[45]. This method leads to an overall lower total sum of distances at the cost of making some non-optimal local assignments. Global minimisation methods are implemented in the most popular tracking softwares Trackmate[23] and UTrack[25].

All tracking methods virtually share one main parameter: the maximum frame-to-frame distance allowed for linking two spots. This parameter limits the number of possible successors of each spot by preventing linking outside the possible motion range of the particle which depends on the intrinsic speed of the imaged particle and the imaging framerate. As it restricts linking, this parameter should ideally be the smallest possible but larger than the valid motion range of the particle.

Another family of methods for tracking are based on Multiple Hypothesis Testing (MHT) which relies on knowing a priori the different modes of motion of the particles and performing simultaneous Bayesian optimisation of the motion parameters and tracking[46]. These algorithms, although being theoretically superior, are problematic in practice as the imprecise characterisation of particle motion, which in the crowded and interaction-prone cellular environment can be spatiotemporally very variable, can bias the generated tracks.

**Table 1 | Extracted diffusion parameters**

| Experiment | Struct-aware | A [95% CI] | $D_1$ (µm²/s) [95% CI] | B [95% CI] | $D_2$ (µm²/s) [95% CI] | $R^2$ | n (disps.) | m (recs.) |
|---|---|---|---|---|---|---|---|---|
| Halo$^{Mito}$ | No | 0.0036 [0.0033, 0.0039] | 0.33 [0.30, 0.35] | 0.0059 [0.0056, 0.0062] | 1.91 [1.77, 2.06] | 0.991 | 13725 | 1 |
| | Yes | 0.0034 [0.0030, 0.0037] | 0.30 [0.27, 0.33] | 0.0061 [0.0058, 0.0065] | 1.69 [1.55, 1.83] | 0.989 | 9103 | 1 |
| TOMM20::Halo | No | 0.0054 [0.0052, 0.0057] | 0.15 [0.15, 0.16] | 0.0044 [0.0041, 0.0046] | 0.55 [0.52, 0.58] | 0.999 | 47822 | 1 |
| | Yes | 0.0055 [0.0053, 0.0058] | 0.15 [0.15, 0.16] | 0.0043 [0.0040, 0.0045] | 0.53 [0.50, 0.55] | 0.999 | 35606 | 1 |
| SEC61B::Halo + BSA | No | 0.0041 [0.0038-0.0044] | 0.27 [0.26-0.29] | 0.0057 [0.0054-0.0059] | 1.14 [1.08-1.21] | 0.998 | 430596 | 7 |
| | Yes + ambig. Removal | 0.0040 [0.0037, 0.0043] | 0.26 [0.25, 0.27] | 0.0058 [0.0055, 0.0061] | 1.07 [1.02, 1.13] | 0.999 | 371482 | 7 |
| SEC61B::Halo + Oleic Acid | No | 0.0039 [0.0036-0.0042] | 0.30 [0.28-0.32] | 0.0059 [0.0056-0.0062] | 1.30 [1.22-1.37] | 0.998 | 403889 | 6 |
| | Yes + ambig. Removal | 0.0038 [0.0035, 0.0041] | 0.29 [0.27, 0.31] | 0.0060 [0.0057, 0.0063] | 1.23 [1.17, 1.30] | 0.998 | 349450 | 6 |

Population contributions and diffusion coefficients of each population fitted from a mixture of 2 Rayleigh distributions (see Methods section "Fitting of displacement lengths distributions") for the data presented in Fig. 4f, g. $n$ is the number of trajectory displacements pooled over the m performed recordings, fits values are given with their 95% confidence intervals.

**Tracking difficulty increases with characteristic lengths.** As shown in our results, the choice of the maximum linking distance is important, especially when increasing spot density. Choosing it too small leads to truncation of the intrinsic displacement distribution of the particles while choosing it too large results in increased linking errors. Theoretically, this parameter can be derived from the intrinsic displacement distribution of the particle at a given framerate. However, this distribution (Rayleigh distribution, Eq. 2) exhibits an increasingly fatter tail with characteristic lengths which widens the distribution towards higher displacement values. This spread can be quantified from the quantile function of the Rayleigh distribution:

$$q(p, \sigma) = \sigma \sqrt{-4\ln(1-p)},$$

with $\sigma = \sqrt{D\Delta t}$ the characteristic length and $p$ the probability. Taking an extreme value for $p$ to encompass most of the distribution, say $p = 0.999$ we get:

$$q(0.999, \sigma) \approx 5.26\sigma.$$

At the same time, the mean of the distribution is given by

$$E = \sigma \sqrt{\pi} \approx 1.77\sigma.$$

Thus, as the characteristic length increases, the maximum linking distance required to cover 99.9% of the displacement distribution increases ~2.97-fold faster than the average displacement value (Fig. S1a). Hence, tracking faster particles require a disproportionately higher linking distance compared to slower particles to capture the tail of the displacement distribution. At the same time, increasing the linking distance increases tracking error and reduces the tolerable spot density. As such, reliable tracking becomes increasingly difficult as the speed of the particle and / or acquisition time increases.

**Tracking from simulations.** Given simulated ground truth data, we reconstruct tracked trajectories using a custom Python script interfacing with Trackmate[23] inside ImageJ[47]. This script loads only the spots from a simulation file and runs the "Simple LAP tracker" algorithm of Trackmate (bypassing spot detection) using the tracking parameters presented in Table 2.

The maximum linking distance value of 6 µm was chosen as it is larger than the 99.9 quantile value of the Rayleigh distribution for the maximum tested characteristic length $l = 1$ µm, $q(0.999, 1) \approx 5.3$ µm.

The nearest neighbour tracking results presented in Fig. S1c were obtained using the corresponding algorithm in Trackmate. We generated the furthest neighbour tracking results (Fig. S1d) by modifying the nearest neighbour algorithm in Trackmate to pick the furthest neighbour instead of the closest. The u-track results (Fig. S1e) were obtained using u-track v.2.3[25] in MATLAB, bypassing spot detection by loading simulated spots, and using the function *trackCloseGapsKalmanSparse* to perform the tracking. In u-track, we used the *costMatRandomDirectedSwitchingMotionLink* cost matrix setting the *maxSearchRadius* parameter to the desired maximum linking distance and disabled gaps by setting the *timeWindow* gap parameter to 0.

**Simulations setup.** To evaluate tracking fidelity as a function of the characteristic length, spot density and maximum linking distance, we generated an ensemble of standardised simulation scenarios. We used a field of view of 420×420 pixels with a pixel size of 0.02419 *µm* leading to a 10.16 × 10.16 *µm* region. Inside this region, we devise the simulations to maintain a constant specified number of spots per frame. When a trajectory ends at some frame, a new trajectory is started at the next frame with the coordinates of its initial point uniformly distributed in the field of view. For all simulations, the distribution of the trajectory sizes followed a Poisson distribution with parameter $\lambda = 13$

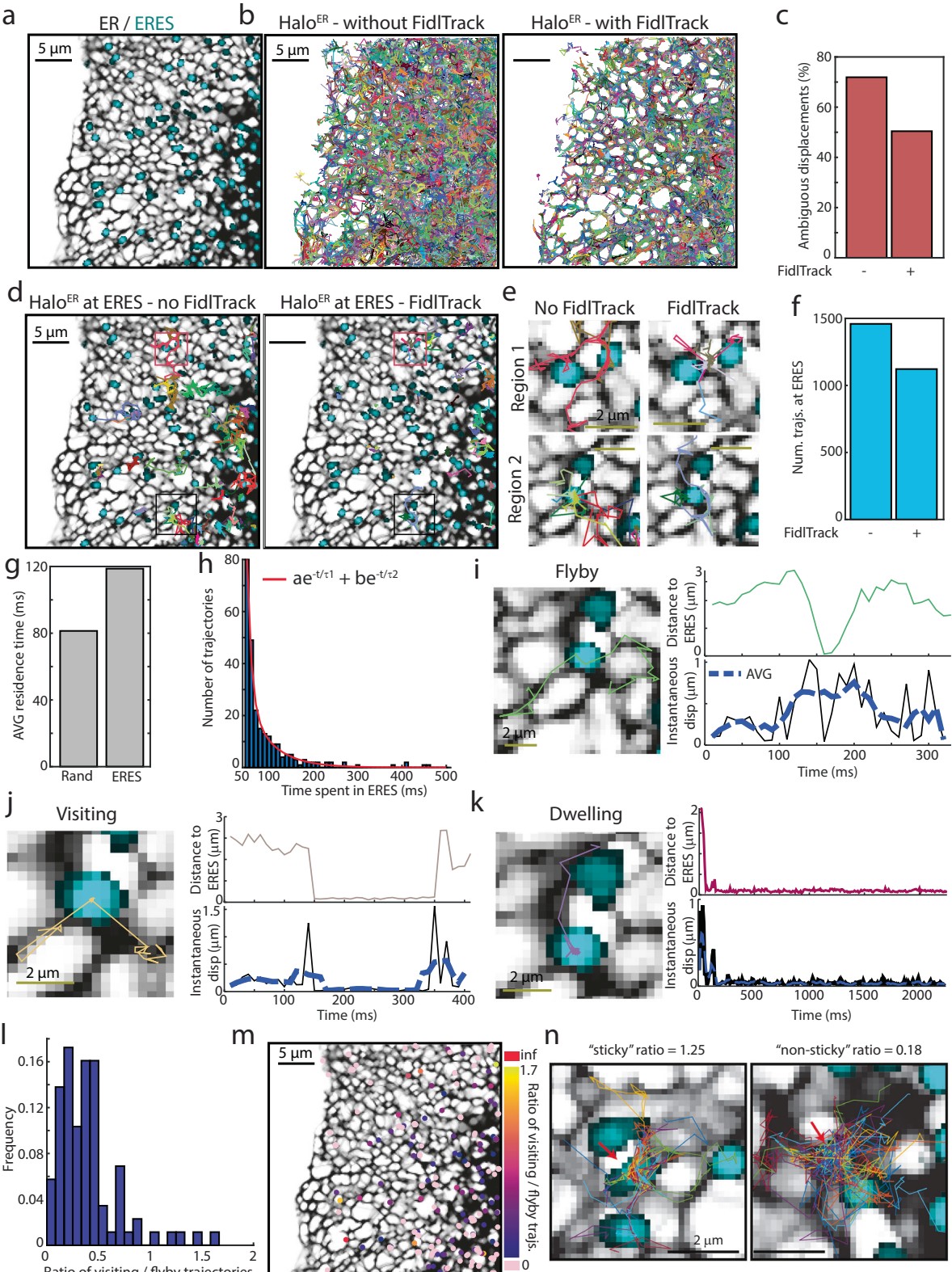

points (Fig. S1i) mimicking what we would generally observe in our SPT recordings. For all simulations, we used the simulation timestep $\delta t = 0.0001$ s and the diffusion coefficient $D = 1 \ \mu m^2/s$. The two remaining parameters were varied as presented in Table 3.

The acquisition timestep was varied so that the characteristic length $l = \sqrt{\Delta t}$ (as $D = 1 \mu m^2/s$) would be equal to $l = \{0.01, 0.0245, 0.05, 0.0748, 0.1, 0.1249, 0.25, 0.3821, 0.5, 0.625, 0.75, 0.875, 1\}$ μm.

The number of simulated frames was adapted to keep the number of displacements similar between simulations such that the number of frames × number of spots per frame = 60000 (e.g., 60,000 frames at 1 spot per frame, 12,000 frames at 50 spots per frame). For the freespace case, each simulation scenario – the combination of acquisition time and number of spots per frame – was independently repeated 5 times.

**Fig. 5 | ER luminal particle behaviour at ER Exit Sites (ERES). a** Time-averaged TIRF images of a Hela cell expressing a SEC61B::GFP ER marker (grayscale) with knock-in SEC13::SNAP (stained with cpSNAP-JF549) ER exit site marker (cyan). **b** Reconstructed Halo$^{ER}$ (ER-targeted HaloTag with KDEL retention signal, stained with PA-JF646) trajectories (individually colour-coded) reconstructed without (left) or with FidlTrack (right). **c** Percentage of ambiguous displacements for the trajectories reconstructed without or with FidlTrack. **d** Trajectories (individually colour-coded) visiting an ERES (spending at least 10 frames in ERES) found without (left) or with FidlTrack (right) overlaid on top of the averaged ER structure (grayscale) and ERES positions (cyan). **e** Blow up on the two regions highlighted in (**d**) showing how ERES act as attractors to local trajectories without FidlTrack (left), a problem mostly corrected when using FidlTrack (right). **f** Number of trajectories at ERES (spending at least 2 points close to an ERES) without and with FidlTrack. **g**, Average amount of time spent by trajectories close to ERES versus close to

random ERES-like sites (see Fig. S6a, b). **h** Amount of the time spent in ERES by trajectories spending at least 5 frames at an ERES with a cutoff at 500 ms. The red line corresponds to a fit of the distribution to a biexponential function of parameters $a = 156$, $\tau 1 = 10$ ms, $b = 29$, $\tau 2 = 56$ ms and $R^2 = 0.997$. **i** Example of a trajectory flying-by (passing without stopping) an ERES (left) and quantification of its distance to the ERES (right, top) and instantaneous displacement length (right, bottom) showing no sign of association with the ERES. **j** Same as (**i**) but for a trajectory visiting for some frames and then exiting an ERES. **k** Same as (**j**) but for a trajectory dwelling for a long time in an ERES (until bleaching). **l** Ratio of visiting to flyby trajectories per exit site. **m** Exit sites colour-coded by their visiting to flyby trajectory ratio as presented in (**l**). **n** Example of two exit sites exhibiting different behaviours, the left one is "sticky" with a high ratio of visiting trajectories while the right one has mostly flyby trajectories. Source data are provided as a Source Data file.

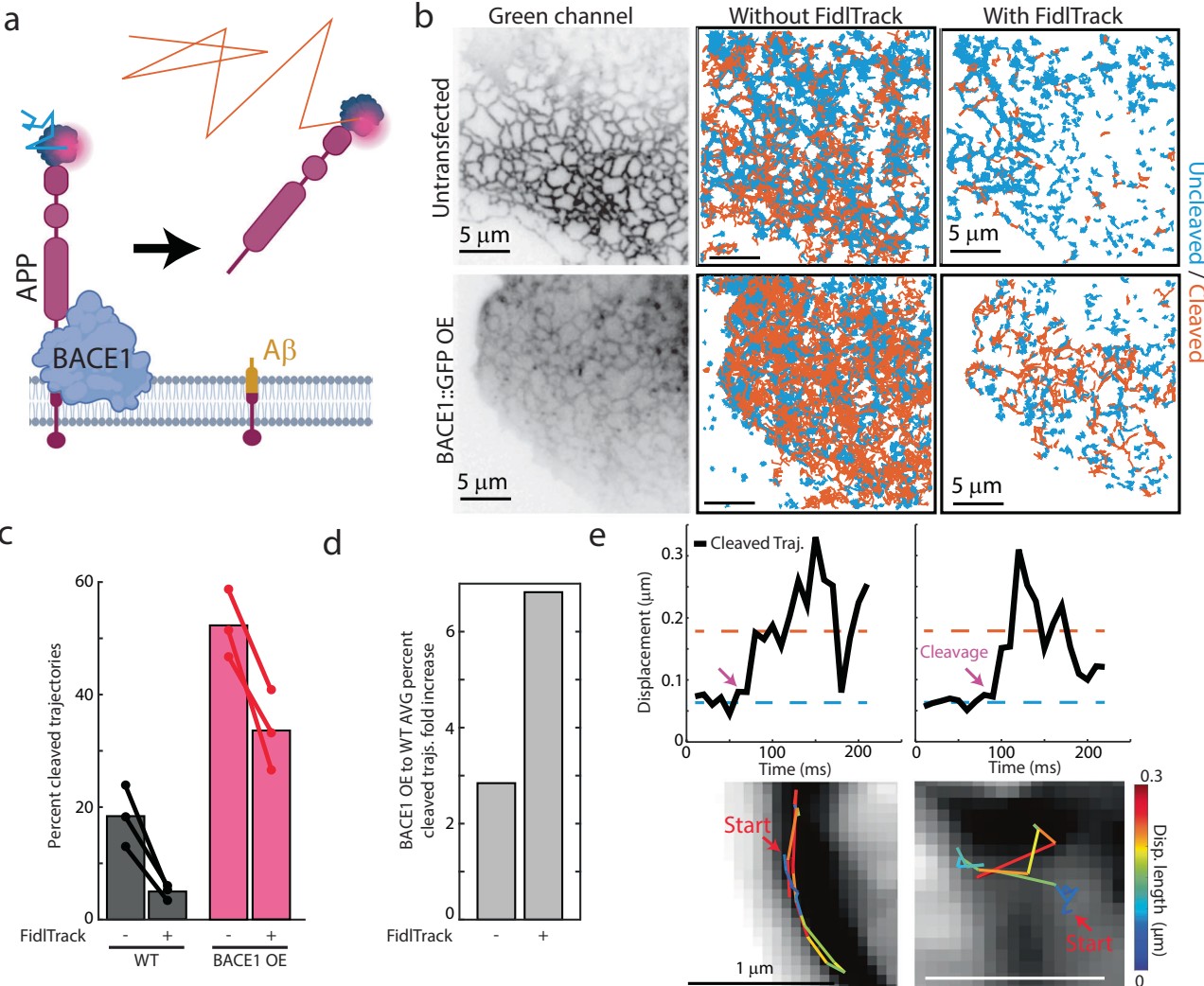

**Fig. 6 | Visualising amyloidogenic APP cleavage in live cells. a** Sketch depicting the principle of the APP cleavage detection assay at the single-molecule level: before cleavage, the N-terminus of APP has a slow membrane dynamics while it acquires a fast luminal dynamics after BACE1 cleavage, created with Biorender[59]. **b** Representative COS-7 cells expressing the ER retention and marker streptavidin::StayGold$^{ER}$ without (top) or with (bottom) BACE1-GFP overexpression as shown by the fuzzy signal in the green channel (left column). Right four panels show trajectories of the N-terminus tagged APP-HaloTag (stained with PA-JF646) reconstructed without (middle column) or with (right column) FidlTrack. Trajectories are colour-coded depending on whether they were classified as cleaved (orange) or uncleaved (cyan) based on their average trajectory displacement length. **c** Barplot of the average percentage of cleaved trajectories for three WT

cells ($n = 3$, left, grey) and three BACE1 overexpressing cells ($n = 3$, right, pink) reconstructed without or with FidlTrack. **d** Barplot of the fold increase of average percent cleaved trajectories from BACE1 overexpression to WT (from (**c**)) without (left) or with (right) FidlTrack reconstruction. **e** Example of detected cleavage events in trajectories reconstructed with FidlTrack from a BACE1-overexpressing cell. The top row presents the instantaneous displacement length along the trajectories (smoothed with a 3-points moving average) with the pink arrow showing the cleavage point. The horizontal dashed lines represent the average displacement for the cleaved (orange) and uncleaved (cyan) populations. The bottom row shows the trajectories overlaid on the local ER structure and colour-coded by displacement length. Source data are provided as a Source Data file.

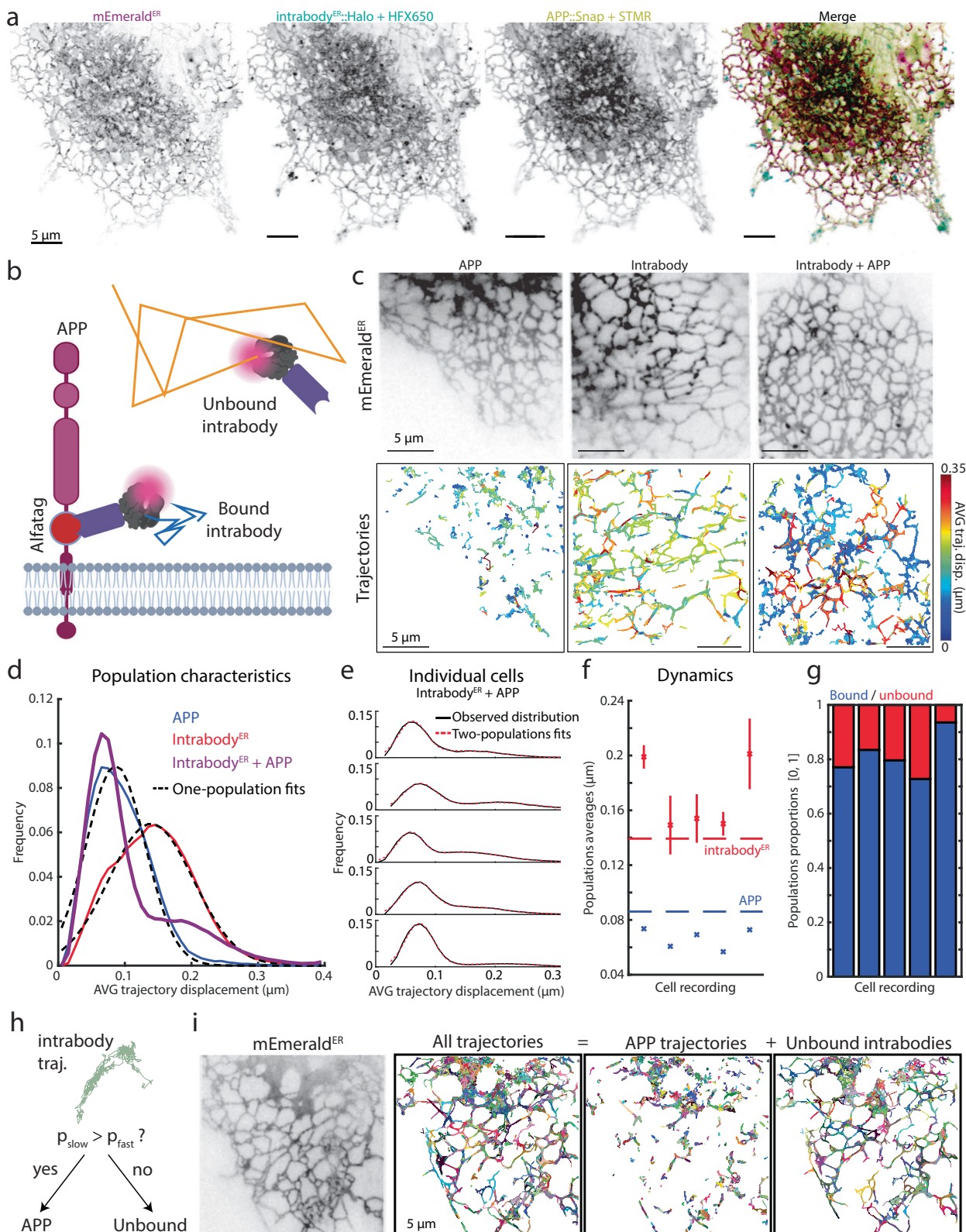

**Comparing ground truth and reconstructed trajectories.** To compare ground truth and reconstructed trajectories, we first matched each spot from the ground truth trajectories $X_{GT}$ to its corresponding spot in the reconstructed trajectories $X_R$ if any. Two spots are the same if they appear at the same frame and within 1e-5 $\mu m$ of each other, in which case we write $m(X_{GT}) = X_R$. If the ground truth spot does not appear in the reconstructed trajectories (it happens when a spot is not appear in the reconstructed trajectories (it happens when a spot is not part of any trajectory and gets removed in the tracking process) then $m(X_{GT}) = \varnothing$.

Based on this matching, a displacement between two spots $X_{GT}^{(1)} \rightarrow X_{GT}^{(2)}$ from a ground truth trajectory is considered incorrect if: at least one spot is not mapped $m(X_{GT}^{(1)}) = \varnothing \lor m(X_{GT}^{(2)}) = \varnothing$ or the mapped spots do not belong to the same trajectory: $id(m(X_{GT}^{(1)})) \neq id(m(X_{GT}^{(2)}))$, with $id$ the function giving the trajectory identifier associated with a spot.

**Fig. 7 | Detecting individual protein status from their dynamics. a** Confocal images of a COS-7 cell stably expressing the ER marker mEmerald[ER], an ER-targeted anti-alfatag intrabody fused to an HaloTag (stained with Halo-JFX650), an Amyloid Precursor Protein containing the Alfatag peptide and fused to a SnapTag (stained with Snap-TMR) and the merged image showing the near perfect correlation of the three signals. **b** Principle of the intrabody status detection method at the single-molecule level. Unbound intrabodies have a relatively fast luminal dynamic while bound intrabodies acquire the slower membrane dynamics of their APP target (Created with Biorender[60]). **c** TIRF images of ER structures (top) and associated FidlTrack reconstructed single-particle trajectories (colour-coded by the average trajectory displacement length of each trajectory) for cells expressing: the intrabody together with the ER-retained Snap-tagged APP (stained with Snap-PA-JF646, left), the ER-retained intrabody alone (stained with Halo-PA-JF646) (middle), or the intrabody (stained with Halo-PA-JF646) together with the APP construct (right). **d** Distributions of the pooled averaged trajectory displacement lengths for the unbound intrabody (red), APP (blue) and intrabody + APP (purple) recordings. The

dashed lines correspond to fits of the distributions to single-components Gaussian models. The pooling is over 5 recordings for each condition. **e** Distributions of the averaged trajectory displacement lengths for 5 individual recordings of the intrabody in the presence of APP. The dashed lines correspond to fits of the distributions to two-component Gaussian models. **f** Average trajectory displacement length parameter (fitted value ± 95% CI) of each component extracted from the fits of the different distributions presented in (**e**). The dashed lines correspond to the average trajectory displacement length parameters of the single-component Gaussian fits of the APP (blue) and intrabody-only (red) recordings from (**d**). **g** Associated proportion of bound and unbound intrabody populations for the fits presented in (**e**) (see Methods section "Binding status extraction from intrabody trajectories"). **h** Principle of the procedure for unmixing APP-bound and unbound intrabody trajectories. **i** Unmixing of APP vs unbound intrabodies on a recording, from left to right: ER structure of a cell expressing both the intrabody and APP, recorded trajectories, APP trajectories and unbound trajectories, colour coded by individual trajectory. Source data are provided as a Source Data file.

## Table 2 | Tracking parameters for simulated data

| Parameter | Values |
|---|---|
| Framegap | 0 |
| Maximum linking distance ($\mu m$) | {0.2,0.4,0.6,0.8,1.0,1.2,1.4,1.6,1.8,2.0,2.2,2.4,2.6,2.8,3.0,3.2,3.4,3.6,3.8,4.0,4.2,4.4,4.6,4.8,5.0,5.2,5.4,5.6,5.8,6.0} |

Framegap and maximum linking distance parameters used to evaluate tracking fidelity.

## Table 3 | Simulation parameters

| Parameter | Values |
|---|---|
| Acquisition timestep $\Delta t$ (s) | 0.0001, 0.0006, 0.0025, 0.0056, 0.01, 0.0156, 0.0625, 0.146, 0.25, 0.3906, 0.5625, 0.7656, 1 |
| Number of spots per frame | 1, 5, 10, 15, 20, 25, 30, 35, 40, 45, 50 |

Ranges of values for the acquisition timestep and number of spots per frame used in the simulations.

The final number of erroneous displacements $N_{err}$ corresponds to the number of mismatched ground truth displacements $N_{mismatch}$ plus the difference between the number of displacements in recovered $N_R$ and ground truth $N_{GT}$ trajectories: $N_{new} = N_R - N_{GT}$, if positive:

$$N_{err} = N_{mismatch} + \max(0, N_{new})$$

This second elements considers any new displacements appearing only in reconstructed trajectories and thus erroneous. Finally, the Fidelity Score corresponds to the fraction of erroneous displacements $f_{err}$ obtained by dividing $N_{err}$ by the maximum number of displacements between the ground truth and reconstructed trajectories:

$$f_{err} = \frac{N_{err}}{\max(N_{GT}, N_R)}.$$

**Fractional Brownian motion simulations.** Given a diffusion coefficient $D$ ($\mu m^2/s$), a value of Hurst exponent $H \in [0, 1]$, an elementary timestep $\delta t$ (sec) and a number of steps $N$, we generate a Fractional Brownian motion trajectory by generating a sequence of $N$ fractional Gaussian noises $B_H^{(k)}$, with $0 \leq k < N$ (using the `fractional_gaussian_noise` from the `stochastic` package in Python). Then, the trajectory is obtained in a way similar to Eq. 1 but replacing the Gaussian noise by the fractional Gaussian noise

$$X(k\delta t) = X((k-1)\delta t) + \sqrt{2D\delta t}B_H^{(k)},$$

With $0 \leq k < N$. The anomalous exponents reported in Fig. S1f, g corresponds to $\alpha = 2H$. The simulation setup and parameters were kept the same as for the Brownian simulations (see Methods section "Simulations setup"). The frame-to-frame displacement length

distributions and population-averaged Mean Squared Displacements curves derived from fractional Brownian motions with $\alpha = \{0.5, 1, 1.5\}$ are presented in Fig. S1j.

**Evaluating the density response of an SPT experiment.** We call density response of an SPT experiment the evolution of the density of correctly or non-ambiguously recovered displacements as a function of the density of spots. This value quantifies the gain in "correct" (non-erroneous or non-ambiguous) trajectory data (displacements) produced by an increase in the amount of input data (spots). Observing the density response curves (Fig. 1h, j, l) for increasing characteristic lengths, we can see a shift from a linear relationship to a saturated one. This means that there exists a maximum tolerable density for high-characteristic lengths setups, where increasing spot density beyond this point results in reconstructing more incorrect than correct displacements. Incidentally, this defines the spatial resolution for the experiment, as this density defines the maximum reachable spatial resolution. Replacing simulation-based error detection by the data-based ambiguity detection makes the saturation phenomenon appear at much lower densities and for a wider range of characteristic lengths (Fig. 2f and Fig. S3f), further reducing the spatial resolution.

To better visualise the evolution of this response as a function of the characteristic length, we fitted a linear function (fit function from MATLAB) to the first four points of each density response curve and extracted its slope. We can see that in all geometries, using ambiguity instead of error detection greatly reduces the density tolerance of the setup and that using graph ambiguity instead of Euclidean distance improves the response, particularly in the case of mitochondria (Fig. S4d).

**Extracting masks from fluorescent images.** To obtain binary structure masks from fluorescence images of an organelle marker, a pre-processing step is performed for the TOMM20 mitochondria dataset (Fig. 4) to improve image quality, applying an unsharp mask with a 20-pixel radius and 0.6 weight in ImageJ. In the case of an image stack (simultaneous SPT and organelle marker recording), a local temporal average over 17 frames (or 51 frames for the intrabody application) is performed to improve contrast. Then the images or stacks are used to train Ilastik[28] Pixel Segmentation classifiers. The training is done by

manually annotating regions belonging to the structure and background along the stack of images until a satisfying segmentation is obtained (as assessed by overlaying the mask and fluorescence images). Post-processing of the mask stack is sometimes applied using closure and dilation operators from ImageJ, to remove isolated groups of mask pixels and make the mask sharper.

### Simulation in organelles geometries

**Extracting masks from fluorescent images and extracting the contour geometry.** To generate the organelle masks for the simulations, we selected 6 (resp. 5) images of ER (resp. mitochondria) networks from peripheral regions of COS-7 cells from the simultaneous SPT and marker recording datasets (such as presented in Fig. 4a). These recordings are 15000 frames long over $128 \times 128$ pixels fields of views with a pixel size of $0.0967821\,\mu m$ and were pre-processed as in Methods section "Extracting masks from fluorescent images". To capture finer geometrical details in these images, we then applied the MUSICAL super-resolution algorithm (MusiJ[48] plugin in ImageJ) to decrease the pixel size of the image by a factor 4. This plugin was run on a sub-stack of 50 frames per recording (to preserve structure locality) with parameters: threshold=4, alpha=6 and "Subpixels per pixel" = 4. The resulting stack of images was average-projected, its contrast enhanced (0.3% saturated pixels) and normalised ("Enhance contrast" in ImageJ) and a Gaussian blur of radius 1.5 pixels was applied. Finally, the images were cropped to keep only the artefact-free central region, resulting in a $420 \times 420$ pixels image with a pixel size of $0.0241955\,\mu m$ (Fig. S2c box 2). These super-resolved images were then segmented as described in Methods section "Extracting masks from fluorescent images" to obtain the binary organelle masks (Fig. S2c box 3).

We then approximated organelles' geometries as the polygonal contour of the masks. We generated contour polygons from mask images using *potrace* v1.16 (available at https://potrace.sourceforge. net/) with the parameter "alphamax" = 0. These polygons consisted of one or multiple compound polygon with a main polygon representing the outline of the geometry with possibly multiple negative polygons representing holes in it. These polygons were extracted from the SVG images generated by *potrace* and fed to our simulation programme (Fig. S2c box 4).

**Brownian simulations in geometries.** To simulate Brownian motion in a confined geometry $G$, given by an ensemble of polygons, we modify Eq. 3 as follows:

$$X(t+\delta t) = \begin{cases} X(t)+Y & \text{if } [X(t), X(t)+Y] \cap G = \varnothing \\ \text{reflect}(X(t), X(t)+Y, G) & \text{otherwise} \end{cases}, \quad (5)$$

where $Y = \sqrt{2D\delta t}\eta$ is a 2D vector of Brownian displacements (see Methods section "Brownian simulations in free space"), $[X(t), X(t)+Y] \cap G$ means the intersection between the segment $[X(t), X(t)+Y]$ and any segment composing the polygon $G$, and *reflect* is an algorithm that performs the reflection of the displacement against polygon boundaries as presented in Box 1.

---

**BOX 1:**

# Algorithm for reflecting a trajectory on the geometry boundaries

```
reflect(X: Point, X': Point, G: geometry)  →  Point
        s = closest_intersecting_segment(X, X', G)
        while s ≠ ∅
                X, X' = reflect_one(X, X', s)
                s = closest_intersecting_segment(X, X', G)
        return X'

closest_intersecting_segment(X: Point, X': Point, G: geometry)  →  Segment
        cur_seg, cur_dist = ∅, +∞
        for s in G
                p = intersection_point(Segment(X, X'), s)
                if p exists and distance(X, p) < cur_dist
                        cur_seg, cur_dist, = s, distance(X, p)
        return s

reflect_one(X: point, X': point, s: Segment)  →  (Point, Point)
        ip = intersection_point(Segment(X, X'), s)
        n = inward_pointing_normal_vector(s)
        s = dot_product(n, X' − ip)
        return (ip, X' − 2sn)
```

When a collision with a segment from the geometry is detected, the algorithm reflects the destination point according to the normal vector of the intersecting segment. When a trajectory segment intersects with multiple polygon segments, the segment with the intersection point closest to the starting point of the trajectory segment is processed. This procedure is repeated until no more intersection with the geometry is found. For this method to be efficient, a small simulation timestep is required so that the number of collisions needed to reposition a displacement is kept low. To keep geometries simple, we did not use any smoothing on the obtained polygons.

The simulations in geometries were conducted in the same way as described for freespace in Methods section "Simulations setup". Simulations were repeated independently 5 times over each of 6 ER and 5 mitochondria geometries (Fig. S2b).

**Ambiguity scoring.** We call ambiguous any displacement for which more than one possible successor (spots within the linking distance in the next frame) spot exists. To detect ambiguous displacements, we rely on the ensemble of trajectories $T$ and the ensemble of detected spots $S$. The two can vary after filtering trajectories on sizes where spots from too-short trajectories get discarded, hence the ensemble of detected spots is required to obtain the true ambiguity. We identify spots by their spatiotemporal coordinates $s = (f, X)$ with $f$ is the frame of appearance and $X = (x, y)$ their 2D coordinates. For an ensemble of $N$ trajectories each possessing $M_i$ points, $T_i = \{(f_{i,j}, X_{i,j})\}$ $(i = 1..N, j = 1..M_i)$, an ensemble of $K$ spots $S = \{(f_k, X_k)\}$ $k = 1..K$, and a maximum linking distance $l$, the ensemble of ambiguous displacements is defined as

$$A(T, S, l) = \left\{ i, j \, \middle| \, \mathrm{card}\left( \left\{ k \, \middle| \, f_k = f_{i,j} + 1 \wedge d(X_{i,j}, X_k) < l, k = 1..K \right\} \right) > 1, i = 1..N, j = 1..(M_i - 1) \right\},$$

where card($E$) is the cardinality of ensemble $E$ and $\wedge$ is the logical and operator. The ambiguity score $a$ is then calculated as the percentage of ambiguous displacements:

$$a(T, S, l) = \frac{card(A(T, S, l))}{\sum_{i=1..N}(M_i - 1)} \times 100,$$

where the denominator gives the total number of displacements.

**Structure-aware Tracking**
**Principle of structure-aware tracking.** When molecules are confined inside a complex structure, we can use the observed geometry to refine the tracking. Indeed, tracking algorithms use the Euclidean distance between spots, although this is correct for small displacement lengths inside complex geometries, it becomes increasingly incorrect as the characteristic length increases. Instead, at longer displacement lengths, a more correct metric is the distance along the structure. We approximate this distance by the length of the shortest path in the graph formed over the connected pixels of the mask image. We consider the nodes of the graph to be the centres of the mask pixels, and we use 3×3 connectivity (including diagonal pixels). The shortest path is computed by the Dijkstra's algorithm[44]. As the spatial resolution of the graph is the pixel size of the mask image, much larger than spatial spot resolution, this can lead to discrepancies with the Euclidean distance especially for short displacement. To counteract that, given a pair of spots located at positions $X, X'$ associated to pixels $p, p'$, we do not consider directly their shortest path $P^*(p, p')$ but the shortest path amongst all their neighbours plus the Euclidean distance between each spot and the selected neighbour pixel centre:

$$d_G(X, X') = d_E(X, p_1) + d\left(P^*(p_1, p_2)\right) + d_E(X', p_2),$$

Where $(p_1, p_2) = \mathrm{argmin}_{q \in nh(p), q' \in nh(p')} d\left(P^*(q, q')\right)$, $nh(p)$ gives all the valid neighbours of pixel $p$, $d_E$ is the Euclidean distance and $d(P)$ is the distance along a path $P$ which corresponds to the sum of the edge lengths (Fig. S4a).

Given that usual SPT recordings are composed of dozens of thousands of points, computing shortest paths in graphs on the fly inside the tracking algorithm is too slow to be of practical use. Instead, we rely on pre-computing the distances between each pair of reachable pixels from the mask, with a user-defined cut-off distance (there is no need to compute graph distances much larger than the targeted maximum linking distance). To keep the resulting file size manageable, we developed a custom binary file format to store the distances. This file is then used by our Trackmate[23] extension where every call to a distance function in the Simple LAP (Jaqaman) tracking algorithm has been replaced by a call to the graph distance.

**Structure-aware Tracking in dynamic structures.** For structures showing high temporal variations, we developed an extension of structure-aware tracking that allows the use of multiple mask images. For a stack of $N$ mask images, we first perform a stabilisation procedure by taking the local maximum value of each pixel over a sliding window of $M$ (typically 5) images. This step reduces local mask variations due to imperfect segmentation. Then, the stack is subsampled into possibly overlapping time windows, keeping for each window the maximum pixel values over the stack images forming this window. Finally, the connected components are computed for each time window and the graph distance between each pair of pixels of each component of each time window is computed.

**Fitting of displacement lengths distributions.** The distribution of displacement lengths for a 2D unconstrained Brownian motion is expected to follow a Rayleigh distribution (Eq. 2). In practice, for the recordings we made on the mitochondria matrix and outer membrane and ER membrane, the displacement distributions were better approximated by a mixture of 2 Rayleigh distributions:

$$f(l) = A \frac{l}{2\sigma_1^2} e^{-\frac{l^2}{4\sigma_1^2}} + B \frac{l}{2\sigma_2^2} e^{-\frac{l^2}{4\sigma_2^2}},$$

with $\sigma_1, \sigma_2$ the parameters of each distribution and $A, B$ their relative contributions. The fits were performed with the fit function from MATLAB, specifying the above custom equation and the following constraints: start point: $A = 0.1$, $\sigma_1 = 0.05$, $B = 0.1$, $\sigma_2 = 0.15$ and $A, B > 0$, $0 < \sigma_1 < M/4$ and $0 < \sigma_2 < M/2$, where M is the maximum bin value (1 for the fits from Fig. 4f and 0.6 for Fig. 4g). The diffusion coefficients are extracted from the values of $\sigma$ and the acquisition time $\Delta t$ as

$$D = \frac{\sigma^2}{2\Delta t}.$$

On simulations, diffusion coefficients were estimated by fitting a Rayleigh distribution to the distribution of displacement lengths (e.g., Fig. S1j) using the maximum likelihood method (raylfit function from MATLAB). To reduce the variability of the estimation, the reported diffusion coefficients in Fig. S3e are obtained by averaging 1000 estimations from randomly selected subsets of 1100 displacements from the total pool of displacements.

**Decoupling particle and organelle motion.** When tracking particles in very mobile organelles, such as mitochondria, the particle motion can be contaminated by the organelle dynamics. To obtain a more precise estimation of the particle motion, we developed a procedure to decorrelate these two sources of motion based on the dynamic masks generated for structure aware tracking. First, we selected a

recording containing a mitochondrion moving simultaneously with the presence of a single-particle spot and spatiotemporally cropped the image stack around this event (Fig. 4i, left). Then, we used the stack of binary images recorded simultaneously with the single-particle data (see Methods section "Extracting masks from fluorescent images") representing the positions of the different mitochondria in the imaged field of view, we label each image to extract the positions of the individual mitochondria and determined their centres. Next, we propagated mitochondria identity between frames through nearest-neighbour tracking between the centres with a cutoff distance of 8.6 pixels. After mitochondria tracking, the convex hull of each mitochondrion at each frame was determined and single-particle trajectories were assigned to a mitochondrion if they fall inside its convex hull. Finally, the centre of mass of the mitochondria mask was computed at each frame forming a trajectory $Y_1, ..., Y_n$ (Fig. 4i, right-top), and its displacement $\Delta_k^{(Y)}$ between each pair of frames $(k, k+1)$ was subtracted to the position of each single-particle spot $X_{k+1}$ observed at frame $k+1$ (Fig. 4i, right-bottom black trajectory)

$$X'_{k+1} = X_{k+1} - \Delta_k^{(Y)},$$

Where $X'$ gives the position of the spot decoupled from the structure's motion (Fig. 4i, right-bottom red trajectory).

**Binding status extraction from intrabody trajectories.** To extract the binding status of an intrabody based on the dynamics of its trajectories, we first acquired control datasets of the intrabody without its target and the target alone. Then, for each recorded trajectory $T = \{X_i\}$, $i = 1..N$, composed of $N_i$ 2D points, we computed its average displacement length as:

$$L(T) = \frac{1}{N} \sum_{i=1}^{N_i-1} ||X_{i+1} - X_i||,$$

Where $||.||$ is the Euclidean norm. We then fitted (using the `fit` function to a "gauss1" model in *MATLAB*) the average displacement lengths distribution of each control to a single-component Gaussian function:

$$g_1(l) = ae^{-\left(\frac{(l-b)}{c}\right)^2},$$

With $a$ a scaling coefficient, $b$ (in μm) the centre of the component and $c$ (in μm) its standard deviation. We then applied a similar procedure to the trajectories of intrabodies in the presence of their target Alfa-tagged APP, but this time using a two-component Gaussian function (fit function to a "gauss2" model in MATLAB):

$$g_2(l) = a_1 e^{-\left(\frac{(l-b_1)}{c_1}\right)^2} + a_2 e^{-\left(\frac{(l-b_2)}{c_2}\right)^2}, \quad (6)$$

With $a_1, a_2$ the scaling coefficients of each component, $b_1, b_2$ their centres (in μm) and $c_1, c_2$ their standard deviations (in μm). For the fit to the individual recordings (Fig. 7e-g), we used the following constraints: $b_1 < 0.1 \mu m$ and $0.1 < b_2 < 0.3 \mu m$ to ensure that the first (resp. second) component fits the slow (resp. fast) population. The fraction $f_k$ of population $k = 1, 2$ (Fig. 7g) is given by: $f_k = \frac{a_k}{a_1 + a_2}$.

**Analysis of the behaviour of ER luminal trajectories at ER exit sites.** The ER mask (Fig. 5a) was obtained from the SEC61B-GFP channel by segmenting the images using Ilastik (see Methods section "Extracting masks from fluorescent images"). The ER exit sites (ERES) displayed in the different Fig. 5 panels correspond to the average ER-exit site image over the entire stack. For Fig. 5f, we consider that a trajectory visits an ERES if it spends more than 10 frames at a distance <0.15 $\mu m$ from the ERES centre at the corresponding frame. The random regions used in Fig. 5g and S6a, b were obtained by selecting 239 (the same number as

ERES) randomly generated points that fall on a pixel with an average mask value > 0.7 (max. 1) and > 1 μm away from any ERES or previously generated centre. The fit presented in Fig. 5h was obtained using the *fit* function from MATLAB using the *exp2* function and the *robust 'LAR'* option. We define a fly-by trajectory as a trajectory spending 2 frames at a distance <0.15 $\mu m$ from the ERES centre at the corresponding frames. For the analysis of ERES behaviours (Fig. 5l-n), only the ERES containing at least 7 different trajectories were analysed.

**Visualising APP cleavage.** N-terminus tagged APP trajectories were considered as being cleaved if their average displacement length is > 0.135 $\mu m$ and uncleaved otherwise (Fig. 6b, c). The percentage of cleaved trajectories (Fig. 6c, Fig. S7a) are obtained by fitting a 2-populations Gaussian function (see Methods section "Binding status extraction from intrabody trajectories") to the distribution of average trajectory displacements using the fit function from MATLAB using the *robust 'LAR'* option and a lower bound for $b_2$ set to 0.15 $\mu m$, and an upper bounds for $b_1$ and $b_2$ set to 0.1 and 0.2 $\mu m$ respectively.

A trajectory was considered as containing a cleavage event if it undergone a rapid shift from a slow to fast dynamics (Fig. 6e) as detected by the following condition: the maximum displacement length <0.6 $\mu m$ and the average of the first 5 displacements is <2 * the average of the last 5 displacements and there is at least one difference between successive displacement $\Delta$ in the range $0.075 < \Delta < 0.5 \mu m$ (the first such difference is denoted $\Delta_1$), and the index $k_1$ at which $\Delta_1$ happens is in the range $5 < k_1 < len(traj) - 10$ and the average of all displacements up to $k_1$ is $< 0.1 \mu m$ and the average of all displacements after $k_1$ is $> 0.15 \mu m$ and the standard deviation of all displacements up to $k_1$ is $\leq 0.03 \mu m$ and the standard deviation of all displacements after $k_1$ is $\geq 0.05 \mu m$ and all displacements are $< 2 *$ the average of all the average trajectory displacements from the cleaved population. This ensemble of conditions ensures that there is a single large transition point in the displacements along a trajectory, and that it is located far enough from both edges to get meaningful estimations of the dynamics before and after the transition.

**Spots localisation precision.** The localisation precision is the error made during the fit of the point spread function to the local maxima of the acquired microscopy image to determine the spots' centres. This error mostly comes from two sources, the pixelation noise and the background noise, and can be estimated by the following formula[49]

$$\langle (\Delta x)^2 \rangle = \frac{s^2 + \frac{a^2}{12}}{N} + \frac{4\sqrt{\pi}s^3 b^2}{aN^2},$$

where $\Delta x$ is the localisation uncertainty, $s$ is the standard deviation of the point spread function, $a$ is the pixel size, $b$ is the intensity of the background noise and $N$ is the number of collected photons. Note that as long as $N$ (controlled by the exposure time of the camera) stays the same, this formula is independent of the acquisition time. Table 4 shows the estimated localisation uncertainty computed using the ImageJ plugin Thunderstorm[50] for the different pixel size and frame-rates combinations used.

In all cases, the localisation uncertainties are <1 pixel which makes unlikely the possibility of falsely positioning a spot outside a mask for the structure-aware tracking process.

In our simulation benchmarks, we considered a constant null localisation noise for all characteristic lengths and densities. This can be explained as the localisation noise can always be kept constant for different framerates by always using the same exposure time (see previous paragraph).

**Experimental protocols**
**Plasmid list.** Table 5 presents the list of plasmids used in the manuscript.

**Table 4 | Spot localisation precision**

| Dataset | Figures | Framerate (Hz) | Pixel size (nm) | Localisation uncertainty AVG ± STD (nm) | N spots |
|---|---|---|---|---|---|
| Halo[ER] | Fig. 2g-k | 50 | 65 | 7.6 ± 2.4 | 996 |
| Halo[Cyto] | Fig. 3f-i | 41.7 | 96 | 18.1 ± 5.3 | 4206 |
| SEC61B | Fig. 4 | 167 | 96 | 18.0 ± 6.6 | 372 |
| Halo[ER] | Fig. 5 | 100 | 159 | 17.1 ± 10.8 | 930 |
| APP (N-term) | Fig. 6 | 100 | 65 | 10.2 ± 3.1 | 1562 |
| Alfatag[ER] | Fig. 7c-g | 167 | 65 | 12.7 ± 4.0 | 259 |

Spot localisation precisions estimated for the different experimental setups used in the manuscript.

**Table 5 | Plasmids**

| ID | Plasmid name | Label | Description | First appearance | Source |
|---|---|---|---|---|---|
| 235 | pLV_CMV_Prss-mEmeraldG-KDEL_IRES_Prss-Halo-KDEL | mEmerald[ER], Halo[ER] | Simultaneous expression of ER lumen targeted mEmerald and HaloTag | Fig. 2g | This study Addgene #251340 |
| 418 | pLV_CMV_mEmerald-KDEL | mEmerald[ER] | ER lumen targeted mEmerald | Fig. 7a | This study Addgene #251341 |
| 123 | pLV-Tet3G | None | Tet-On 3G transactivator protein | Fig. 6b | Addgene #128061 |
| 716 | pLV_CMV_Prss-AtagIB-Halo-KDEL | intrabody[ER] | ER targeted anti-Alfatag intrabody with Halotag | Fig. 7a | This study Addgene #251342 |
| 717 | pLV_Tre3g_Prss-AppAtag-Snapf | APP | APP containing Alfatag and SnapTag | Fig. 7a | This study Addgene #251343 |
| 737 | pLV_Tre3g_Halo | Halo[Cyto] | Cytosolic HaloTag | Fig. 3f | This study Addgene #251344 |
| 267 | CMV_4xmts_HaloTag | Halo[Mito] | Mitochondria matrix targeted HaloTag | Fig. 4a | This study Addgene #251345 |
| 826 | CMV_TOMM20_HaloTag | TOMM20::Halo | Halotag fused to TOMM20 | Fig. 4a | This study (sub-cloned from Addgene #55146) |
| 250 | CMV_4xmts_mNeonGreen | mNeonGreen[Mito] | Mitochondria matrix targeted mNeonGreen | Fig. 4a | Addgene #98876 |
| 827 | mEmeraldMito | mEmerald[Mito] | Mitochondria matrix targeted mEmerald | Fig. 4a | Addgene #54160 |
| 828 | SEC61B::HaloTag | SEC61B::Halo | Halotag fused to SEC61B | Fig. 4a | From[13] |
| 829 | mEmerald-KDEL | mEmerald[ER] | ER lumen targeted mEmerald | Fig. 4a | from[58] |
| 1014 | pcDNA3.1_GFP-SEC61B | SEC61B:: GFP | SEC61B fused to GFP | Fig. 5a | Addgene #121159 |
| 70 | pFLAG_ER Halo | Halo[ER] | ER lumen targeted Halotag | Fig. 5b | Gift from David Ron's lab |
| 466 | BACE1-GFP | BACE1-GFP | BACE1 fused to GFP | Fig. 6b | Addgene #165032 |
| 931 | pLV_Tre3g_SBP_Halo_APP_SNAPf_CMV_SS_Strp_staygold_KDEL | APP-Halo | N-term tagged APP and ER-retained strep-staygold | Fig. 6b | This study Addgene #251346 |

List of the plasmid used in the manuscript.

## Lentiviral particles and stable cell-lines production

A total of $10^6$ Hek293FT (ThermoFischer R70007) cells were seeded in T75 flasks and grown overnight in complete DMEM medium (Sigma-Aldrich #F9665), 1% L-glutamine (Gibco #25030024) and 1% penstrep (Gibco #15140122)). The next day, third-generation viral packaging plasmids (Addgene #12251, #12253, #12259) and the plasmid of interest were transfected using Polyethylenimine (PEI, Polysciences #24765) with a plasmid ratio of 1:1:1:1. The media was then changed every 24 h for three days and collected on days 2 and 3. The collected media was filtered with a 0.45 μm filter (Millipore #SLHA033SS) and collected into 50 ml tubes, 2 ml of 20% sucrose water solution was added at the bottom of each tube which were then centrifugated at $14{,}000 \times g$ for

3h45min at 4 °C. The formed pellet containing the viral particles was resuspended in 30 μl of PBS (Gibco #14190136) per 50 ml of collected media and stored at −80c.

To produce stable cell lines, COS-7 cells (ATCC CRL−1651) were seeded in a 24-well plate in 0.4 ml complete DMEM (DMEM (Sigma-Aldritch #D6429) medium supplemented with 10% FBS (Sigma-Aldrich #F9665), 1% L-glutamine (Gibco #25030024) and 1% penstrep (Gibco #15140122)) and 1 to 10 μl (depending on the particle efficiency) of viral particles were added to the media overnight. The next day, the media was changed to fresh medium, and the cells were allowed to grow. The cells used in the luminal KDEL recordings were made from COS-7 dATL[51] infected with mEmeraldG-KDEL_IRES_Prss-Halo-KDEL (plasmid

235). For the experiments involving APP cleavage, COS-7 cells stably expressing the Tet-On system (plasmid 123) were infected with plasmid 931. For the experiments involving intrabody and APP, the different cell lines used were made sequentially in the following way: COS-7 WT were first infected with the Tet-On system (plasmid 123), then with mEmerald-KDEL (plasmid 418), then with AtagIB-Halo-KDEL (plasmid 716) constituting the "intrabody alone" cell line. Finally, the intrabody + APP cell line was obtained by infecting the "intrabody alone" cell line with AppAtag-Snapf (plasmid 717).

HeLa SEC13-SNAP knock-in cells were generated by transfecting WT Hela CCL-2 (ECACC General Collection) with a pX330 plasmid harbouring a guide RNA and a homology repair plasmid using FuGENE. The SEC13 genomic locus (gene ID 6396) was targeted with the guide RNA: 5'-ACGAGCAGTGACAAGACAGGTGG-3' located after the stop codon in the SEC13 coding sequence. To generate the SEC13-SNAP-V5-PolyA-Puromycinr HR plasmid, the empty SEC13 HR plasmid was linearized with BamHI and EcoRI. The BamHI-SNAP-V5-PolyA-Puromycinr-EcoRI fragment was cut out from a plasmid containing the tag and resistance cassette as previously described[52,53]. For selection of positively-edited cells, puromycin (1 μg/mL) was added to the cells 3 days after transfection.

## Live-cell SPT imaging

Three different microscopes were used for single-molecule imaging with all recordings performed using HILO illumination. The ER-luminal probe, intrabody and APP (Fig. 2g-k and Figs. 6,7) data were acquired on a custom-built wide-field microscope with a 100×1.45 NA TIRF objective (Olympus), a Prime BSI sCMOS (Teledyne, USA) camera and a beam splitter (Cairn Research Ltd., UK) separating two or three wavelengths on a single field of view. The SEC61B, mitochondrial and neuronal data (Fig. 3f-i and Fig. 4) were obtained on an Elyra 7 (Zeiss) TIRF equipped with a 63×1.46 NA TIRF objective, two pco.edge sCMOS cameras (PCO, Germany), and a beam splitter separating two different wavelengths in each camera (Cairn Research Ltd., UK). Both microscopes were equipped with a stage incubator maintaining the cells at 37 with 5% CO2.

The ER exit site data (Fig. 5) were acquired on a Nikon Eclipse Ti N-STORM system enclosed in an incubator chamber set to 37 °C and 5% CO2. Fluorescence excitation was achieved using highly inclined and laminated optical sheet illumination (HILO)[54]; with laser lines at 405 nm, 488 nm, 561 nm, and 647 nm. Light was collected through a Nikon CFI Apo TIRF 100XC oil-immersion objective and directed to a Hamamatsu Orca Flash 4.0 V3 sCMOS camera. Using a single camera, the imaging approach had to be carefully designed to perform 3-colour imaging and (1) avoid crosstalk, (2) achieve a sufficient frame rate to image single-molecules of Halo-KDEL, and (3) maintain decent signal-to-noise ratios for all channels. In order to minimize crosstalk, and without using rotating filters that would compromise the frame rate, cells were labelled so that dim-to-bright channels were organized from longer to shorter emission wavelengths. This way, single Halo-KDEL molecules were imaged at 647 nm excitation, endogenously expressed SEC13-SNAP marker was imaged at 561 nm, and overexpressed GFP-SEC61B marker at 488 nm. The 488 nm laser served also as photoactivation of the PA-JF646-HaloTag ligand dye, so there was no need for further photoactivation using the 405 nm laser. The laser power densities for the 488 nm, 561 nm, and 646 nm channels were measured under epi illumination at the front aperture of the objective and were 0.52 W/cm², 0.71 W/cm² and 34 W/cm², respectively. High frame rates that didn't compromise signal-to-noise ratios of the ER and ERES markers were achieved by reducing the field of view to 200 × 200 pixels (31.8 × 31. 8 μm, pixel size is 159 nm) and combining it with alternating excitation at 200 Hz. Single-molecules images were taken every even frame. ER and ERES markers were imaged in alternating phases of consecutive odd frames. The frames in every phase were then averaged together to create a single image, with improved signal-to-noise ratio. This resulted in an effective frame rate of 100 Hz for the single-molecule channel, and an effective frame rate of 12.5 Hz for the fully labelled ER and ERES masks. Imaging was performed for a total of 56 s per video.

For the recordings presented in Figs. 2–4,6 and 7, the protein of interest was fused to a HaloTag (or SnapTag for APP in Fig. 7) protein and stained with Halo-PA-JF646[7] (or Snap-PA-JF646) photoactivatable ligand (gift from Lavis lab, Janelia Research Campus) and excited with a continuous 646 nm laser (between 30 and 50 mW laser input – 5.8 to 9.9 mW at the coverslip). The density of spots per frame was controlled by adjusting, before each recording, the power of the continuous 405 nm laser (between 0 to 20 mW laser input – 0 to 36 μW at the coverslip). The combination of sparse staining and photoactivable dye, allowed us to achieve fairly constant spot densities across frames. The different staining procedures used are presented in Table 6.

For overnight staining (Figs. 2g-k, 3f-i, 7), cells were stained in 2 ml for COS-7 / 0.4 ml for ineurons, complete DMEM (DMEM (Sigma-Aldritch #D6429) medium supplemented with 10% FBS (Sigma-Aldrich #F9665), 1% L-glutamine (Gibco #25030024) and 1% penstrep (Gibco #15140122)) / iN2 medium (Neurobasal medium (Gibco #21103049) supplemented with 20 μl/ml B-27 (Gibco #17504044), 2mM L-glutamine (Gibco #25030024), 50 μM 2-mercaptoethanol (Gibco #31350010), 10 μl/ml penstrep (Gibco #15140122), 1ug/ml DOX (Fischer Scientific #BP26531), 10 ng/ml NT-3 (Gibco #450-03–10UG) and 10 ng/ml BDNF (Gibco #10477253)). The next day, the cells were washed 2 times with PBS (Gibco #14190136), left to rest in 2 ml / 0.4 ml of fresh medium for ~15 min, washed 1x with PBS (Gibco #14190136) and the media was replaced with 2 ml / 0.4 ml of imaging medium. For COS-7, the cells were grown in single-well polymer-bottom microscopy dishes (IBDI) and the imaging medium consisted of FluoroBrite DMEM (Gibco #A1896701) medium supplemented with 10% FBS (Sigma-Aldrich #F9665), 1% L-glutamine (Gibco #25030024) and 1% penstrep (Gibco #15140122). For iNeurons, cells were grown in 4-well polymer bottom chambers (IBDI) and the imaging medium consisted of IN2 using phenol red-free Neurobasal medium (Gibco #12348017).

For short-pulse staining (Fig. 4), cells were grown on glass coverslips and 18 h post-transfection, cells were washed once with 3 ml PBS (Gibco #14190136) before incubation with 0.5 μM PA-JF646 in optiMEM (Gibco #31985062) for 1 min. Cells were then washed 5x with 10 ml of PBS (Gibco #14190136), followed by one wash with 10 ml of complete culture medium.

For the ER exit sites recordings (Fig. 5), cells were cultured in phenol red-free DMEM (Capricorn Scientific, #DMEM-HXRXA), supplemented with 10% v/v FBS (Capricorn Scientific, #FBS-HI–11A), 2 mM L-glutamine (Biowest, #X0550–100), and 100 U/mL penicillin and 100 μg/mL streptomycin (Biowest, #L0022-100). A day before transfection, cells were seeded on μ-slide 8 well glass bottom chambers (Ibidi, #80827). Transfection was performed using X-tremeGENE 9

**Table 6 | Staining procedures**

| Dye | Staining | Target | First appearance |
|---|---|---|---|
| Halo PA-JF646 | 75 nM, overnight | Halo[ER] | Fig. 2g |
| Halo PA-JF646 | 25 nM, overnight | Halo[cyto] | Fig. 3f |
| Halo PA-JF646 | 0.5 μM, 1 min | Halo[Mito] | Fig. 4a |
| Halo PA-JF646 | 0.5 μM, 1 min | TOMM20::Halo | Fig. 4a |
| Halo PA-JF646 | 0.5 μM, 1 min | SEC61B::Halo | Fig. 4a |
| cpSNAP JF549 | 0.25 μM, 15 min | SEC13::Snap | Fig. 5a |
| Halo PA-JF646 | 1 μM, 15 min | Halo[ER] | Fig. 5b |
| Halo PA-JF646 | 5 μM, 30 min | APP::Halo | Fig. 6b |
| Halo PA-JF646 | 75 nM, overnight | intrabody[ER]::Halo | Fig. 7c |
| Snap PA-JF646 | 1.5 μM, 1 h | APP::Snap | Fig. 7c |

Staining procedures used in the different SPT recordings presented in the manuscript.

**Table 7 | Tracking parameters for experimental data**

| Dataset | Figures | Framerate (Hz) | Spot radius (µm) | Spot threshold | Max. linking distance (µm) | Min. traj. length |
|---|---|---|---|---|---|---|
| Halo^ER | Fig. 2g-k | 50 | 0.55 | 0.3 | 2.8 (conventional) | 15 |
| Halo^Cyto | Fig. 3f-i | 41.7 | 0.75 | 0.2 | 6 (conventional & struct-aware) | 5 |
| Halo^Mito | Fig. 4a-f | 167 | 0.75 | 1.1 | 1 (struct-aware) | 10 |
| TOMM20 | Fig. 4a-f | 167 | 0.75 | 1 | 0.6 (struct-aware) | 10 |
| SEC61B | Fig. 4a-f | 167 | 0.75 | 1 | 0.6 (struct-aware) | 5 |
| SEC61B (BSA / OA) | Fig. 4g,h | 167 | 0.75 | 1 | 1 (struct-aware) | 5 |
| Halo^ER | Fig. 5 | 100 | 0.75 | 6 | 2 (conventional & struct-aware) | 10 |
| APP (N-term) | Fig. 6 | 100 | 0.6 | 0.5 | 1 (conventional & struct-aware) | 20 |
| Alfatag^ER | Fig. 7 | 167 | 0.7 | 0.35 | 1 (struct-aware) | 5 |
| APP | Fig. 7 | 167 | 0.7 | 0.35 | 1 (struct-aware) | 5 |

Spot radii and detection threshold are used in the spot detection process, the maximum linking distance is used in the tracking and the minimum trajectory length is a post-processing step.

(Roche, #6365779001) at a 3:1 reagent (µL) to plasmid (µg) ratio, according to the manufacturer's protocol. The amount of plasmid added per cm² of culture area was 66 ng of Halo-KDEL and 33 ng of GFP-SEC61B. Imaging was performed from 16 h to 24 h post-transfection. To clean unbound dye, samples were washed multiple times with growth media: starting with three washes, continued by a minimum 2 h incubation at 37 °C and 5% $CO_2$, and ended with an additional three washes.

For the APP cleavage recordings (Fig. 6), cells were plated in an 8-well dish, the next day, 500 ng of BACE1-GFP plasmid was transfected in each well using lipofectamine 3000 (Invitrogen) and following the manufacturer's protocol. The cells were images 24 h after transfection.

All trajectories were reconstructed with Trackmate[23], using the "LoG detector" for spot detection and the "simple LAP tracker" for tracking using the parameters presented in Table 7.

The frame gap parameter was always set to 0.

The spot detection threshold is an intensity threshold used in the LOG detector algorithm to separate genuine spots from background.

**Fluorescence recovery after photobleaching (FRAP) recordings**
FRAP experiments were performed on a scanning confocal LSM 780 (Zeiss) fitted with a Plan-Apochromat 63x/1.40 Oil DIC M27 objective (Zeiss). The samples were maintained at 37 °C in a humidified imaging chamber with 5% $CO_2$. For each recording, a circular region of radius ~3.5 µm was selected in the peripheral region of a cell containing a well distinguishable ER network. The evolution of the fluorescence signal was recorded at 0.6 Hz and photobleaching was performed inside the circular region for 1.6 s (at frame 6) by setting the 561 nm laser power to 50% maximum power. The recordings were processed in imageJ[47] with the FRAP profiler v2 plugin (Hardin lab) based on the bleaching ROI provided in the recording files. The experiments used SEC61B-HaloTag stained with Halo-TMR (Promega).

**Confocal imaging**
The images of cellular expression of APP and the intrabody (Fig. 7a) were obtained using a STELLARIS8 (Leica, Wetzlar, Germany) confocal microscope with on-stage incubation maintaining cells at 37 °C and 5% $CO_2$. For this experiment, the cells were grown at low density on a glass coverslip and stained overnight with Halo-JFX650 (25 µM) and with Snap TMR (NEB #S9105S, 500 µM) for 1 h. Following TMR staining, the cells were washed 2x with PBS (Gibco #14190136), left to rest for 30 min in the incubator, washed 1x with PBS (Gibco #14190136), transferred to an imaging chamber and the medium was changed to FluoroBrite DMEM (Gibco #A1896701) medium supplemented with 10% FBS (Sigma-Aldrich #F9665), 1% L-glutamine (Gibco #25030024) and 1% penstrep (Gibco #15140122). The imaging setup was as follows: excitation lasers: 488 nm at 0.1% intensity, 555 nm at 5% intensity and 650 nm at 5% intensity and the emission detectors were set at: mEmerald: 493–548 nm (50 gain), TMR: 596–631 nm (90 gain) and, JFX650 704-834 nm (90 gain). The pinhole was set to 1 AU and a line averaging of 7 was used.

**Induced pluripotent stem cells maintenance and differentiation**
Human Induced Pluripotent Stem Cells (hiPSCs) with stable integration into a safe-harbour locus of the NGN2 transgene under a doxycycline-inducible promoter (i3 neurons[29], from Michael E. Ward, National Institute of Health), were cultured in TESR E8 medium (STEMCELL technologies #05990). Differentiation was induced by replacing the culture medium with DMEM/F12 (Gibco #10565018) supplemented with 10 µl/ml N-2 supplement (Gibco #17502048), 2mM L-glutamine (Gibco #25030024), 10 µl/ml NEAA (Gibco #11140035), 50 µM 2-mercaptoethanol (Gibco #31350010), 10 µl/ml Pen-Strep (Gibco #15140122) and 1 µg/ml DOX (Fischer Scientific #BP26531). This medium was changed daily for two days. On day 3, the medium was replaced to an iN2 neuronal maintenance Neurobasal medium (Gibco #21103049) supplemented with 20 µl/ml B-27 (Gibco #17504044), 2mM L-glutamine (Gibco #25030024), 50 µM 2-mercaptoethanol (Gibco #31350010), 10 µl/ml penstrep (Gibco #15140122), 1ug/ml DOX (Fischer Scientific #BP26531), 10 ng/ml NT-3 (Gibco #450-03-10UG) and 10 ng/ml BDNF (Gibco #10477253). At day 4, cells were transferred to a polymer bottom 4-well chamber dish (IBDI #80416) coated with poly-L-lysine (Bio-Techne # 3438-100-01) and laminin (Merk #L2020-1MG, 1:100 dilution). When transferred, the media was supplemented with 3 µM uridine (Sigma-Aldrich #U3003-5G), 1 µM 5-Fluoro-2′-deoxyuridine (Sigma-Aldrich #F0503-100MG), and 10 µM rock inhibitor (BD biosciences #562822) for 24 h. The maintenance medium was subsequently changed every day. One day before imaging, the Neurobasal from the maintenance medium was replaced with Neurobasal minus phenol red medium (Gibco #12348017).

**Statistics and reproducibility**
All presented micrographs are representative of at least three independent cell cultures. Relevant source data are provided in the 'Source Data' file.

**Reporting summary**
Further information on research design is available in the Nature Portfolio Reporting Summary linked to this article.

## Data availability
The simulated trajectories data generated in this study have been deposited in the Figshare database under accession code https://doi.org/10.6084/m9.figshare.30940613. The raw trajectories and FRAP

data generated in this study have been deposited in the Figshare database under accession code https://doi.org/10.6084/m9.figshare.30943214. Any other data is available from the corresponding authors upon request. Plasmids generated for this study are available on Addgene (https://www.addgene.org/Edward_Avezov). Source data are provided with this paper.

## Code availability

The codes used to develop the simulations, perform the analyses and generate results in this study are publicly available and have been deposited in the following repositories: SPT_fidelity at https://github.com/pparutto/SPT_fidelity; FidlTrack at https://github.com/Avezovlab/FidlTrack; and structsptsim at https://github.com/pparutto/structsptsim, under MIT license. The specific versions of the codes associated with this publication are archived in Figshare[55–57].

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

## Acknowledgements

We are extremely grateful to Luke D. Lavis (HHMI Janelia Research Campus, USA) for the Janelia Fluor dyes, Junjie Hu (Chinese Academy of Sciences, China) for the ATL KO COS-7 cell line, and Matthew J. Gratian and Mark Bowen (CIMR microscopy facility, UK) for technical support. This work is supported by the UK Dementia Research Institute award UK DRI-2004 through UK DRI Ltd, principally funded by the Medical Research Council. E.A. was further supported by Evelyn Trust and Alzheimer's Society grant AS-525 (AS-PhD-19a-015). RPL, MFGP, and FC acknowledge support by the Government of Spain, State Research Agency (AEI) PID2023-147711NB-100 and PID2022 – 138282NB-I00 project funded by the MCIN/AEI/10.13039/501100011033/FEDER,UE, Fundació CELLEX (Barcelona), Fundació Privada Mir-Puig, the Generalitat de Catalunya through the CERCA programme and AGAUR (Grant No. 2021 SGR01450), and the European Commission H2020 Programme under grant agreement ERC Adv788546 (NANO-MEMEC). This work was partially funded by CEX2024-001490-S [MICIU/AEI/10.13039/501100011033]. F.C. is also supported by "Unidad de Excelencia María de Maeztu" CEX2024-001431-M, funded by MICIU/AEI/10.13039/501100011033. JEC is funded by the Medical Research Council (Ref: MCMB MR/Y011813/1). J.N.A. was supported by funding from a Sir Henry Wellcome award (218651/Z/19/Z) and Career development award (227745/Z/23/Z) from the Wellcome Trust. The Zeiss Elyra7 was supported by a Wellcome Trust equipment bid (212892/Z/18/Z).

## Author contributions

P.P., J.N.A., J.E.C. and E.A., designed the research; P.P., Y.Y., V.D., R.P.L., J.E.C. and J.N.A. performed the experiments; S.E., F.B. and K.G. contributed original reagents and cell lines; P.P. analysed the data, developed and performed the simulations; P.P., J.N.A., J.E.C. and E.A. conceived the project; F.B., M.F.G., F.C., and C.F.K. supervised microscopy work; P.P. and E.A. supervised the project and wrote the original manuscript with inputs from J.N.A. and J.E.C.; all authors discussed and revised the paper.

## Competing interests

The authors declare no competing interests.

## Additional information

[1]UK Dementia Research Institute at the University of Cambridge, Cambridge, UK. [2]Department of Clinical Neurosciences, School of Clinical Medicine, University of Cambridge, Cambridge, UK. [3]Department of Chemical Engineering and Biotechnology, University of Cambridge, Cambridge, UK. [4]ICFO-Institut de Ciencies Fotoniques, The Barcelona Institute of Science and Technology, Barcelona, Spain. [5]Institute of Chemistry and Biochemistry, Freie Universität Berlin, Berlin, Germany. [6]Institució Catalana de Recerca i Estudis Avançats (ICREA), Barcelona, Spain. [7]Department of Medicine and Life Sciences (MELIS), Universitat Pompeu Fabra (UPF), Barcelona, Spain. [8]Cambridge Institute for Medical Research (CIMR), Department of Medicine, University of Cambridge, Cambridge, UK. [9]Cambridge Institute for Medical Research (CIMR), Department of Clinical Neurosciences, University of Cambridge, Cambridge, UK. [10]These authors contributed equally: Joseph E. Chambers, Jonathon Nixon-Abell. ✉e-mail: pvp23@cam.ac.uk; ea347@cam.ac.uk

