## [Transparent Peer Review file · Nature Communications]

FidTrack: high-fidelity structure-aware single particle tracking resolves intracellular molecular motion in organelles sensing APP processing

Corresponding Author: Dr Pierre parutto

Version 0:

Reviewer comments:

Reviewer #1

(Remarks to the Author)

The authors introduced FidTrack for single particle tracking, utilizing information from organelle structures to constrain particle tracking algorithms, thereby producing results with higher fidelity. The manuscript is generally well presented. However, I do not support its publication, at least in its current form, due to the following aspects.

Methodology Novelty:

Numerous tracking algorithms have been developed over the past decades. The manuscript does not clearly articulate what is novel about FidTrack. For simulations, a previously developed data generator was used. For tracking, the authors encountered the same dilemma when multiple spots were within the linking distance, leading to ambiguous tracks. While the ambiguity score can improve overall fidelity, it sacrifices a significant portion of trajectory segments. Structure-aware tracking is a promising feature, but its application has not been fully exploited. Additionally, under the guidance of structural information, tracking has been constrained to very simple scenarios, such as 2D linear or confined motion at low molecule density, which do not necessitate sophisticated tracking algorithms.

The authors should conduct a quantitative and fair comparison against several widely used tracking algorithms (such as those cited in reference 21 or references 23, 25, 36, 37, 38) using both simulated and experimental data to demonstrate the performance and novelty of FidTrack.

Biological Application:

Tracking in organelles such as the ER or mitochondria is a promising direction. Under these conditions, the motion of the molecules is the cumulative result of the organelle's motion and the motion of the molecules within the organelle. It would be useful to separate the motion of the molecules from that of the organelle to obtain more accurate tracking results.

The supplementary videos (such as Videos S3, S6, S7, S8, S9–S11) show a relatively low density of single molecules. As mentioned above, the confined motion and low density make tracking a relatively simple task. The authors should demonstrate the performance of FidTrack under more complex conditions. Ideally, the authors should illustrate biological applications where FidTrack outperforms other algorithms.

(Remarks on code availability)

Reviewer #2

(Remarks to the Author)

This is an important manuscript that will undoubtedly help democratize the use of single particle tracking in cell biology, particularly when studying dynamics in complex geometries, such as the membranes of mitochondria or the ER. The open access tools that accompany this manuscript are an important asset to this manuscript. In general, I think that the paper is very clear and written in a very didactic and systematic manner. I have a number comments for the authors, which I hope will help them provide an even stronger manuscript that will resonate better to the community. I know the list may seem long, but they are all aiming at trying to improve the paper even further. In addition, I think most of them could be responded to easily with some clarifications in the rebuttal and/or revised text. From my point of view, the most important points to clarify are #1,

6, 10, 12, 19, 20, 24.

- 1) Line 92: I agree that the use of the characteristic length $l = \sqrt{D/f}$ is a good choice (for Brownian motion) to avoid increasing complexity in the number of parameters that are to be adjusted for an SPT experiments. However, I wonder whether, from a purely experimental point of view, all "l" values are equivalent, that is for instance same l with different frame rates (and accordingly different D values) can lead to different localization precision. I think it will be interesting for the authors to further discuss on these aspects.
- 2) Line 115: what is the effect of the ROI size on the point being made there?
- 3) Line 141 (Fig 2d): maybe I missed it, but how does this removal affect the trajectory length distribution?
- 4) Line 149-152: maybe it'd be interesting to discuss how to deal with this anisotropic distribution of ambiguous displacements.
- 5) Line 171: the removal of impossible traj. segments depends, I guess, on the accuracy of the segmentation. Can the authors discuss on that?
- 6) Line 173: I agree that there's an improved tracking fidelity, but the improve is very modest (up to 5% I think based on the results in Fig. 3e). Could the authors state this and discuss?
- 7) Line 180-181: I think this statement is confusing and potentially misleading for the inattentive reader. Instead of (or in addition to) the density of recovered non-ambiguous displacements, wouldn't it be more logical to compute the correct vs. incorrect links?
- 8) Line 193: is the illumination widefield or HILO (or TIRF)? Not sure I got it in the methods section either.
- 9) Line 218-219 (and Fig. 5a): Maybe it's my poor eyes, but I think I see the opposite in the figure (intrabody ER + punctate, APP mostly ER). Is there a problem with the image labelling?
- 10) Discussion: A general limitation, which is well stated in the text, is that the movement of the particles is assumed to be Brownian. However, as the authors are well aware, this is rarely the case inside cells. I understand this might be beyond the scope of this article, but if they'd consider an extension to other types of motion, I think this would even increase further the importance of this manuscript and associated tools.
- 11) Line 332-333: Is there a factor 4 inside the sqrt (as in the equation) or 2 (as stated in the text)?
- 12) Line 348-250: the single molecule localization precision is not included in the algorithm. How could this affect the results (e.g., a molecule could be localized outside a structure by position, but within the margin of the loc. precision it could fall inside the structure). Could the authors further discuss whether this plays an important effect or if this is considered somehow in the algorithms?
- 13) Line 358: the simulations are kept at a constant density of spots per frame. How does this compare to the experiments? How is this controlled in the experimental situation? If this assumption did not apply in the experiments, how would that affect the results from the algorithms?
- 14) Line 361: I think it'd help to show, as a supp. figure, the data on the traj. length distribution.
- 15) Line 377-377: how often (what fraction of spots) does this occur?
- 16) Line 411: what does "possibly" mean here? Could the authors be more precise?
- 17) Line 416: same here: what does "satisfying segmentation" mean? Could the authors quantify?
- 18) Section 1.4.1: This is a bit cumbersome to understand only by the text, I think that a supplementary figure that explains this with some schemes will help the readers understand this method faster.
- 19) Eq. 3: What happens if $X(t) + Y$ belongs to G but the line connecting the two points crosses the boundaries of the geometry G (that is, it first gets out of G and reenters G, so it eventually ends up being still in G)?
- 20) Line 463: how is the normal vector defined for a polygonal structure? Is there some smoothing used? otherwise the trajectory could show false reflections by boundaries that are polygonal.
- 21) Line 471: could the authors quantify this point? E.g., by computing the average number of reflections per displacement steps?
- 22) Lines 501-508: also here, I think that a supplementary scheme explaining this algorithm would help the readers.
- 23) Line 591: Can the authors provide more details about the photoactivation laser? Is it pulsed or continuous illumination? Also, what is the range of powers? Similar for excitation parameters?
- 24) Line 601-604: how is the protocol for SNAP ligands?
- 25) Line 606 (table): What is the "Spot threshold" and why is this important?
- 26) Figure 1c: maybe having the bar of the characteristic length in log scale will help seeing the small "l" values.
- 27) Figure 1e: legend is missing (I guess it's the same as in f, but specify please)
- 28) Fig. 2a: Do the authors know or have an idea why the behavior of the ambiguity error as a function of the char. length is not monotonous?
- 29) Fig. 5g: How do these results depend on intrabody:APP stoichiometry?

(Remarks on code availability)

I don't have the knowledge that allows me to review the code.

Reviewer #3

(Remarks to the Author)

This work presents an SPT optimization method called FidlTrack, which improves trajectory tracking accuracy by recognizing organelle structures and optimizes tracking parameters through Brownian motion simulation data. Additionally, the authors propose an ambiguity scoring method that identifies and removes ambiguous displacements, thereby enhancing tracking accuracy. These contributions provide some value in improving the quality of trajectory tracking in intracellular diffusion dynamics studies. However, the applicability and generalizability of the method still need further validation, especially in more complex cellular environments. There are clear limitations in some aspects of the study, such as the lack of consideration for diffusion coefficient heterogeneity and the potential data loss during ambiguity removal.

The following points are raised for consideration:

1. Brownian Motion Simulation for Parameter Optimization: The study generates SPT data using Brownian motion simulations to determine experimental and tracking parameter settings for maximum reliability. However, the model assumes a uniform diffusion coefficient. In reality, the diffusion coefficient within the cell is likely to be distributed and may vary in different locations and times. The diffusion process in biological environments can be influenced by various factors such as obstacles, protein aggregation, and the complexity of the intracellular environment. The current model does not account for this heterogeneity, which could limit the generalizability of the simulation results. It would be beneficial to simulate anomalous diffusion trajectories with heterogeneous properties to see whether this would affect the parameter settings.

2. Structure-Aware Tracking: The study employs a structure-aware tracking method, which effectively excludes implausible trajectories by recognizing organelle structures, reducing errors in trajectory identification, and considering the geometric characteristics of organelles and local displacements. This is a commendable approach. It improves tracking accuracy, especially for organelles with clear and stable structures (e.g., mitochondria and the endoplasmic reticulum). However, this method may be less applicable to other organelles or more complex intracellular environments (such as the cytoplasm). If the tracking target is influenced by a complex intracellular environment (e.g., heterogeneous cytoplasmic environment, the effectiveness of the structure-aware method may be compromised.

3. Ambiguity Scoring and Data Loss: By introducing ambiguity scoring and removing ambiguous displacements, the authors effectively reduce errors arising from multiple matching possibilities, thereby improving tracking accuracy. The segmentation of trajectories successfully eliminates ambiguity, enhancing the reliability of the final trajectories. However, ambiguity removal results in data loss. The manuscript does not sufficiently discuss whether this data loss impacts the accurate representation of diffusion dynamics, especially in highly complex dynamic environments. It is recommended to include experimental data to analyze whether removing ambiguity affects key diffusion dynamics.

4. Comparative Analysis with Traditional Methods: Although the authors highlight the advantages of structure-aware tracking and ambiguity removal, the comparative analysis with traditional methods is insufficient. What are the performance differences between structure-aware methods and traditional methods under various conditions (e.g., different organelles, experimental setups)? The attempt to use FidlTrack to detect the binding of antibodies and the Amyloid Precursor Protein (APP) in the ER lumen is interesting but lacks enough analysis. Why is the displacement distribution a key factor for distinguishing binding? Why is the traditional method unable to detect this? It would be helpful to provide additional analysis or experiments to demonstrate the advantages of FidlTrack over traditional methods.

(Remarks on code availability)

Version 1:

Reviewer comments:

Reviewer #1

(Remarks to the Author)

The authors have adequately addressed my concerns, and the manuscript is now suitable for publication.

(Remarks on code availability)

Reviewer #3

(Remarks to the Author)

I thank the authors for addressing my concerns. The revisions have improved the paper to a certain extent. I support its publication.

(Remarks on code availability)

Reviewer #4

(Remarks to the Author)

Acting in position of the original reviewer #2, I found that the authors have addressed the questions in a thorough and satisfactory manner.

(Remarks on code availability)

The instructions have provided sufficient details for running the code, although the reviewer has not attempted to run the code and test the output with actual data.

We thank the editor and the reviewers for their constructive commentary to our manuscript and for specific ideas for improving, testing and demonstrating the utility of our approach to intracellular single molecule tracking. We have now added new study cases demonstrating the resolving power of our techniques and addressed all the comments as detailed below. The original comments are in black font, our responses in blue, and the added/modified text in the revised paper in orange font.

Reviewer #1 (Remarks to the Author):

Reviewer 1

Methodology Novelty:

1) Numerous tracking algorithms have been developed over the past decades. The manuscript does not clearly articulate what is novel about FidlTrack. For simulations, a previously developed data generator was used.

We welcome the opportunity to better articulate the critical novelty of our approach and more vividly showcase its power to reveal new biology.

It is important to note that FidlTrack is not a tracking algorithm. Instead, it is an approach to critically improve the fidelity and accuracy of SPT datasets obtained with broadly used tracking algorithms. We agree that this novelty was not clearly communicated in our original manuscript. FidlTrack provides an unprecedented enhancement in the accuracy of SPT datasets, which we argue is necessary for reliably tracking intracellular biology, especially where it is represented by rare molecular events. To illustrate this, we have generated two new study cases that explore engagement of luminal proteins at ER exit sites (ERES) (**new Fig. 5, added text page 6 paragraphs 5 and Methods section 1.10 page 18, videos S12**), and ER-based amyloid precursor protein (APP) cleavage (**new Fig. 6, added text page 7 paragraphs 2 & 3 and Methods section 1.11 page 18, videos S13-14**) – both of which are rare intracellular events. We also emphasise across the introduction, results and discussion the novelty of our structural aware tracking and ambiguity analysis, and the critical advantages they afford.

It is also important to note that the data generators used for our simulations were originally developed for this study (existing tools being inadequate for the purpose). All our codes and software are publicly available as tools for generating ground truth and analysing fidelity for end-users.

2) For tracking, the authors encountered the same dilemma when multiple spots were within the linking distance, leading to ambiguous tracks. While the ambiguity score can improve overall fidelity, it sacrifices a significant portion of trajectory segments. Structure-aware tracking is a promising feature, but its application has not been fully exploited. Additionally, under the guidance of structural information, tracking has been constrained to very simple scenarios, such as 2D linear or confined motion at low molecule density, which do not necessitate sophisticated tracking algorithms.

The reviewer is correct, the fidelity improvement from ambiguity filters out a portion of trajectories. This is however relatively easy to compensate for with longer acquisitions. Thus, we argue that it can be considered a worthwhile sacrifice for increasing data reliability. We also thank the reviewer for appreciating the value of structure awareness and agree that applying this new

approach in challenging scenarios would better demonstrate its significance, where conventional SPT fails.

The applicability of these two modules is now presented in the two new study cases showing how FidlTrack resolves events that would be massively impacted by trajectory errors without it: visualisation of molecules entry to the early secretory pathway exit sites, and detection of beta secretase amyloidogenic activity (**new Fig. 5 & 6, added text page 6 paragraphs 5 & page 7 paragraphs 2 & 3 and Methods sections 1.10 and 1.11 page 18, videos S12-14**).

3) The authors should conduct a quantitative and fair comparison against several widely used tracking algorithms (such as those cited in reference 21 or references 23, 25, 36, 37, 38) using both simulated and experimental data to demonstrate the performance and novelty of FidlTrack.

We have now expanded our comparisons to include three other tracking algorithms: nearest-neighbour – which is commonly used in tracking, furthest neighbour – covering the worst-case scenario, and u-track – another commonly used tracking algorithm advertised to be efficient at high particle densities. These are presented in the **new panels Figure S1c-e** and described in the **added text on page 4 paragraph 1** and to **Methods section 1.1.7 page 13**. These demonstrate that the gains from our methodology transfer to other tracking algorithms.

Biological Application:

4) Tracking in organelles such as the ER or mitochondria is a promising direction. Under these conditions, the motion of the molecules is the cumulative result of the organelle's motion and the motion of the molecules within the organelle. It would be useful to separate the motion of the molecules from that of the organelle to obtain more accurate tracking results.

We thank the reviewer for this very useful suggestion, which we were able to implement from the segmentation used for structure-aware tracking. We have now introduced a feature to decouple the tracked particles and organelle motions. We show an example in the **new panel Fig. 4i** and **new video S11** where we decouple the motion of TOMM20 molecules from the mitochondrion on which they reside. The new data and technique are described in the **added text on page 6 paragraph 4** and **Methods section 1.8 page 17**.

It is also worth noting that the motion of large structures is often an order of magnitude slower than single molecules, contributing negligibly to SPT. In these cases, the above unmixing is not required.

5) The supplementary videos (such as Videos S3, S6, S7, S8, S9–S11) show a relatively low density of single molecules. As mentioned above, the confined motion and low density make tracking a relatively simple task. The authors should demonstrate the performance of FidlTrack under more complex conditions. Ideally, the authors should illustrate biological applications where FidlTrack outperforms other algorithms.

The study cases we added (**new Fig. 5 & 6, added text page 6 paragraphs 5 & page 7 paragraphs 2 & 3 and Methods sections 1.10 and 1.11 page 18, videos S12-14**) represent challenging conditions, necessitating high-density tracking due to the transient and rare nature of the events of interest (entry to ERES and amyloidogenic BACE1 activity, please see also response to point 2).

The study cases demonstrate, as per reviewers' suggestion, the performance of FidlTrack under complex condition, revealing key events otherwise inaccessible.

Reviewer #2 (Remarks to the Author):

This is an important manuscript that will undoubtedly help democratize the use of single particle tracking in cell biology, particularly when studying dynamics in complex geometries, such as the membranes of mitochondria or the ER. The open access tools that accompany this manuscript are an important asset to this manuscript. In general, I think that the paper is very clear and written in a very didactic and systematic manner. I have a number comments for the authors, which I hope will help them provide an even stronger manuscript that will resonate better to the community. I know the list may seem long, but they are all aiming at trying to improve the paper even further. In addition, I think most of them could be responded to easily with some clarifications in the rebuttal and/or revised text. From my point of view, the most important points to clarify are #1, 6, 10, 12, 19, 20, 24.

1) Line 92: I agree that the use of the characteristic length $l = \sqrt{D/f}$ is a good choice (for Brownian motion) to avoid increasing complexity in the number of parameters that are to be adjusted for an SPT experiments. However, I wonder whether, from a purely experimental point of view, all "l" values are equivalent, that is for instance same l with different frame rates (and accordingly different D values) can lead to different localization precision. I think it will be interesting for the authors to further discuss on these aspects.

We thank the reviewer for drawing attention to this point. We have now clarified that the localisation precision can be kept constant by keeping the exposure time fixed across different framerates on the camera. The frame rates need to be adjusted to D (see Fig. 1c) to obtain the appropriate l and to neutralise any contribution from D to its value. See **added texts on page 3 paragraph 2** and **Methods page 19 section 1.12**.

2) Line 115: what is the effect of the ROI size on the point being made there?

The ROI size will create boundary effects, preventing the observation of large displacements from spots appearing near the edge of the field of view. We were mindful of this in our simulations and selected an ROI size large enough compared with the displacement length to ensure this has a very limited effect, now clarified on the **added text page 4 paragraph 3**.

3) Line 141 (Fig 2d): maybe I missed it, but how does this removal affect the trajectory length distribution?

Ambiguity removal shortens trajectories, as it eliminates dubious connection that may merge more than one particle into single trajectory. We have now added the **new panel Fig. S3d** and **added text on page 4 paragraph 4**, discussing the distribution of trajectory lengths before and after ambiguity removal. Note that the reconstructed trajectories (Fig. S3d left) are longer than in the ground truth as different trajectories get erroneously linked together without application of ambiguity removal.

4) Line 149-152: maybe it'd be interesting to discuss how to deal with this anisotropic distribution of ambiguous displacements.

The ambiguity map (Figure 4j) is a convenient way to deal with anisotropic ambiguity distribution. It can be used to detect and selectively exclude trajectories in problematic regions. We have now made this point explicit in the **added text on page 5 paragraph 2**.

5) Line 171: the removal of impossible traj. segments depends, I guess, on the accuracy of the segmentation. Can the authors discuss on that?

The reviewer is right, it is important to maximise the accuracy of the structure mask as it will influence the extent of improvement afforded by structurally aware SPT. Too loose segmentation reduces the effect of structure-aware tracking, while overly stringent automated segmentation may omit parts of the structure, discarding valid particles and thus reducing the amount of recovered information. We have now added this point to the discussion in the **added text page 8 paragraph 3**.

6) Line 173: I agree that there's an improved tracking fidelity, but the improve is very modest (up to 5% I think based on the results in Fig. 3e). Could the authors state this and discuss?

The reviewer is correct about the extent of improvement for this case. To expand the discussion on how geometry determines the extent of fidelity improvement, we added simulations, modelling conditions with large improvement, and linked this to cases where it plays out biologically. The new simulations in a typical neurites' geometry with structures arranged in parallel (**new Fig. 3c**) shows a case with 50% gain in fidelity (**new Fig. 3d**) and 78% reduction of ambiguity (**new Fig. 3e**). These are reflected in **new and modified texts on page 5 paragraph 5** and discussion in the **added text on page 8 paragraph 3**.

7) Line 180-181: I think this statement is confusing and potentially misleading for the inattentive reader. Instead of (or in addition to) the density of recovered non-ambiguous displacements, wouldn't it be more logical to compute the correct vs. incorrect links?

We agree, we have now **rephrased the sentence on page 5 paragraph 4 / page 6 paragraph 1**. The goal of this panel is to show that these systems exhibit maximum tolerable densities above which no more correct or non-ambiguous displacements are recovered. The plots based on correct links are presented in Fig. 1h.

8) Line 193: is the illumination widefield or HILO (or TIRF)? Not sure I got it in the methods section either.

In **added text to Methods page 22 paragraph 1** we clarify that HILO illumination was used here

9) Line 218-219 (and Fig. 5a): Maybe it's my poor eyes, but I think I see the opposite in the figure (intrabody ER + punctate, APP mostly ER). Is there a problem with the image labelling?

We thank the reviewer for spotting this error, which we have now corrected, see **modified Fig. 7a** (Fig. 5 has now been moved to Fig. 7).

10) Discussion: A general limitation, which is well stated in the text, is that the movement of the particles is assumed to be Brownian. However, as the authors are well aware, this is rarely the

case inside cells. I understand this might be beyond the scope of this article, but if they'd consider an extension to other types of motion, I think this would even increase further the importance of this manuscript and associated tools.

This is an excellent point. In response, we explored the extent of fidelity improvement by FidlTrack for different modes of motion (sub-diffusive, super-diffusive – through Fractional Brownian motion, and mixed Diffusion coefficients) modes. We found that, in agreement with the reviewer's expectation, the approach can optimise fidelity across modes. The new simulations are presented in the **new Fig. S1f-h**, see also **added texts on page 4 Paragraph 1** and **Methods section 1.1.7 page 13**.

11) Line 332-333: Is there a factor 4 inside the sqrt (as in the equation) or 2 (as stated in the text)?

We agree with the reviewer, the text was confusing as a factor 2 was coming from the 2D Brownian motion term, and a factor 2 was already present from the Rayleigh distribution combining to a factor 4. We have now removed the confusing text, **leaving the equation on page 11**.

12) Line 348-250: the single molecule localization precision is not included in the algorithm. How could this affect the results (e.g., a molecule could be localized outside a structure by position, but within the margin of the loc. precision it could fall inside the structure). Could the authors further discuss whether this plays an important effect or if this is considered somehow in the algorithms?

It is an interesting point raised by the reviewer. We have now estimated the localisation error in the different setup we use through the ImageJ plugin Thunderstorm (see **new Table in Methods page 19**, see also response to point 1 above). We found that in all cases localisation precisions are well below 1 pixel (smallest pixel size of the three microscope setups is 65 nm). Considering that both spots and structures are acquired in the same condition, they both suffer from the same diffraction limitations, thus as spots tends to not be exactly localised, structures tend to be larger than they really are. Thus, an error of < 1 pixel makes it unlikely that a spot would be incorrectly considered outside the structure. We have added this discussion to the **new Methods section 1.12 on page 19**.

Importantly, the structure-aware tracking script monitors the number of spots detected outside the structure. In case too many spots get discarded, the segmented mask can be dilated to correct for this problem.

13) Line 358: the simulations are kept at a constant density of spots per frame. How does this compare to the experiments? How is this controlled in the experimental situation? If this assumption did not apply in the experiments, how would that affect the results from the algorithms?

Using a combination of sparse labelling and a photoactivable dye, we achieve a fairly constant spot density across frames (see Reviewer Figure 1). We note that although the average spots densities vary between experiments, their standard deviation remain low indicating a quite stable spot density through time.

Reviewer Figure 1: number of spot per frame in an example recording of Halo^{ER} from Fig. 5 (left) and barplot presenting the average number of spots per frame with standard deviation for the showcase experiments: 1-5: nanobodyER (Fig. 7), 6-10: alfatagged APP (Fig. 7), 11-15: nanobodyER in presence of alfatagged APP (Fig. 7), 16-17: N-term tagged APP (Fig. 6).

In case the spot density is very variable, our results indicate that selecting a slightly larger linking distance is preferable to a slightly smaller one (Fig. 1d). Now clarified this in the **added discussion text page 8 paragraph 2** and **added Methods text page 22 paragraph 3**.

14) Line 361: I think it'd help to show, as a supp. figure, the data on the traj. length distribution.

We have now added the **new panel Fig. S1i** and linked it on the **Methods text page 12 paragraph 4**.

15) Line 377-377: how often (what fraction of spots) does this occur?

The number of spots lost during the tracking process is low as it only consists of spots that are not incorporated in any trajectories (i.e. trajectories of length 1) which are discarded (see Reviewer Figure 2). We made this point clearer in **the added text to Methods page 13 paragraph 1**.

Reviewer figure 2: percentage of simulated spots remaining after tracking (compared to the number of simulated spots) for the density and characteristic length scenarios explored in the main text.

16) Line 411: what does "possibly" mean here? Could the authors be more precise?

We have now **updated the text on Methods page 14 paragraph 2** to specify where this is done.

17) Line 416: same here: what does "satisfying segmentation" mean? Could the authors quantify?

We have now **clarified the text on Methods page 14 paragraph 2** to explain that this step requires visual assessment as the current state of automated segmentation tools limits full automation.

18) Section 1.4.1: This is a bit cumbersome to understand only by the text, I think that a supplementary figure that explains this with some schemes will help the readers understand this method faster.

We have now added a graphical scheme (**new Fig. S3e**) depicting the process and **linked it to the text on Methods page 14 paragraphs 4 & 5**).

19) Eq. 3: What happens if $X(t) + Y$ belongs to G but the line connecting the two points crosses the boundaries of the geometry G (that is, it first gets out of G and reenters G , so it eventually ends up being still in G)?

We thank the reviewer for noting this point. To detect a collision, we do not rely on the destination point $X(t)+Y$ being inside or outside the geometry but rather on whether the $[X(t) X(t)+Y]$ segment intersects with any of the polygon segments. We have now **presented the relevant form of formula 4 on page 14** and **updated the text on Methods pages 15 paragraph 1**.

20) Line 463: how is the normal vector defined for a polygonal structure? Is there some smoothing used? otherwise the trajectory could show false reflections by boundaries that are polygonal.

There is one normal vector for each segment of the polygon, as one needs to keep computational times practical. We felt that no smoothing is needed here as it complicates the geometry, with a significant cost of computation time for little gain. As a quality control step, the simulations include a “stuck detector” that discards a trajectory if the next point cannot be successfully generated in less than 50 trials. This is clarified in the **added text to Methods page 15 paragraph 2**.

21) Line 471: could the authors quantify this point? E.g., by computing the average number of reflections per displacement steps?

At low characteristic lengths, displacements rarely encounter boundaries, whereas when the characteristic length becomes larger than the average confinement length, more collisions occur (reviewer Fig. 3).

Reviewer figure 3: Average number of reflections in ER geometry simulations as a function of the characteristic length.

22) Lines 501-508: also here, I think that a supplementary scheme explaining this algorithm would help the readers.

We have now added panel **Fig. S4d** and **linked to Methods page 16 paragraph 2** explaining the algorithm.

23) Line 591: Can the authors provide more details about the photoactivation laser? Is it pulsed or continuous illumination? Also, what is the range of powers? Similar for excitation parameters?

Our illumination is continuous, and we have now included the information on laser power settings to the **added text in Methods on page 22 paragraphs 2 & 3**.

24) Line 601-604: how is the protocol for SNAP ligands?

SNAP ligand staining procedures are given in the staining **table on pages 22 / 23**.

25) Line 606 (table): What is the "Spot threshold" and why is this important?

This is an initial filtering phase that is commonly used to remove background signal peaks that are not genuine particles. We have now added a sentence describing the spot threshold parameter on the **added text in Methods page 24 paragraph 2**.

26) Figure 1c: maybe having the bar of the characteristic length in log scale will help seeing the small "l" values.

We now added the colour bar to Fig. 1c in log scale.

27) Figure 1e: legend is missing (I guess it's the same as in f, but specify please)

We thank the reviewer for noting this, we have now made the legend visible.

28) Fig. 2a: Do the authors know or have an idea why the behavior of the ambiguity error as a function of the char. length is not monotonous?

This is an interesting observation. The ambiguity-associated error reported in Fig. 2a is computed as the ratio of intra-trajectory errors and the total number of displacements. Increasing characteristic length leads to increased linking distances (as shown in Fig. 1g). This then increases the error at the edges of the trajectory (see **new Fig. S3d** left, showing that trajectories become longer at higher characteristic lengths). Thus, as the characteristic length increases, more boundary errors are committed, reducing the contribution of the ambiguity-associated error. We have added this clarification to **the added text on page 4 paragraph 4**.

29) Fig. 5g: How do these results depend on intrabody:APP stoichiometry?

Stoichiometry will most certainly affect the bound intrabody fraction, reflected in the variability of its quantification (**Fig. 7g, previously Fig. 5g**). We have **updated the text on page 8 paragraph 1** to clarify this point.

As an additional example, the quantifications of the new recording presented in the new **Fig. S7a-c** detects ~57% bound intrabody trajectories.

Reviewer #3 (Remarks to the Author):

This work presents an SPT optimization method called FidTrack, which improves trajectory tracking accuracy by recognizing organelle structures and optimizes tracking parameters through Brownian motion simulation data. Additionally, the authors propose an ambiguity scoring method that identifies and removes ambiguous displacements, thereby enhancing tracking accuracy. These contributions provide some value in improving the quality of trajectory tracking in intracellular diffusion dynamics studies. However, the applicability and generalizability of the method still need further validation, especially in more complex cellular environments. There are clear limitations in some aspects of the study, such as the lack of consideration for diffusion coefficient heterogeneity and the potential data loss during ambiguity removal.

The following points are raised for consideration:

1. Brownian Motion Simulation for Parameter Optimization: The study generates SPT data using Brownian motion simulations to determine experimental and tracking parameter settings for maximum reliability. However, the model assumes a uniform diffusion coefficient. In reality, the diffusion coefficient within the cell is likely to be distributed and may vary in different locations and times. The diffusion process in biological environments can be influenced by various factors such as obstacles, protein aggregation, and the complexity of the intracellular environment. The current model does not account for this heterogeneity, which could limit the generalizability of the simulation results. It would be beneficial to simulate anomalous diffusion trajectories with heterogeneous properties to see whether this would affect the parameter settings.

The reviewer raises an important point, and we are grateful for this suggestion. We have now extended the simulations to cover motion modes beyond simple diffusion. The results, presented in the **new Fig. S1f-h (added text page 4 paragraph 1, discussion page 8 paragraph 2, and Methods section 1.1.7 page 13)**, demonstrate how the methodology allows to optimise fidelity across all modes of motion. As we show in the **new Fig. S1j**, the long-term dynamics does not influence the quality of tracking, only the frame-to-frame displacement distribution is important. If the exact mode of motion cannot be presumed, but the frame-to frame displacement distribution does not obviously deviates from that of a Brownian motion (Rayleigh distribution in 2 dimensions, see Methods section 1.1.2), the best approximation for optimising fidelity is to consider Brownian motion.

2. Structure-Aware Tracking: The study employs a structure-aware tracking method, which effectively excludes implausible trajectories by recognizing organelle structures, reducing errors in trajectory identification, and considering the geometric characteristics of organelles and local displacements. This is a commendable approach. It improves tracking accuracy, especially for organelles with clear and stable structures (e.g., mitochondria and the endoplasmic reticulum). However, this method may be less applicable to other organelles or more complex intracellular environments (such as the cytoplasm). If the tracking target is influenced by a complex intracellular environment (e.g., heterogeneous cytoplasmic environment, the effectiveness of the structure-aware method may be compromised.

We thank the reviewer for recognising the strength of the structure-aware approach. The reviewer is correct, geometrical features of cell components needed to inform the algorithm may not be available for tracking in cytoplasm. Even though the primary goal of the structure-awareness feature of FidTrack is enhanced tracking in structures, its principle extends to cytoplasmic cases

in which it is organised in space, on the appropriate scale. This is exemplified in tracking molecules in the cytoplasm of neurites (Fig. 3f-i). Further, similar principles apply to the tracking of soluble cytoplasmic proteins if the environment can be translated into maskable structural features (e.g. obstacle/aggregate, space occupied by a large organelle etc). Note that, we added a feature allowing decoupling the motion of mobile structures, **new Fig. 4i, added texts on page 6 paragraph 4 and Methods page 17 section 1.8**).

In cases where there is limited structural information, the other features of FidlTrack, optimising the linking distance and removing ambiguities, should still provide a substantial improvement. We have now **modified the discussion page 8 paragraph 4 / page 9 paragraph 1** to reflect these points.

3. Ambiguity Scoring and Data Loss: By introducing ambiguity scoring and removing ambiguous displacements, the authors effectively reduce errors arising from multiple matching possibilities, thereby improving tracking accuracy. The segmentation of trajectories successfully eliminates ambiguity, enhancing the reliability of the final trajectories. However, ambiguity removal results in data loss. The manuscript does not sufficiently discuss whether this data loss impacts the accurate representation of diffusion dynamics, especially in highly complex dynamic environments. It is recommended to include experimental data to analyze whether removing ambiguity affects key diffusion dynamics.

We extended our analyses against available ground truth to show how ambiguity removal affects the results (**new Fig. S3e, added texts page on 4 paragraph 4, Methods page 17 paragraph 3**). We observed that despite removing a substantial amount of data (Fig. S3c), ambiguity removal does not affect the diffusion estimation in cases of medium particle density and characteristic length. This is seen in simulations and in experimental data (**modified Table 1**, now comparing conventional and structure-aware and ambiguity removal tracking, and **added text page 6 paragraph 3**). Critically, in conditions of higher density and characteristic length (top right section of the map Fig. S3e), ambiguity removal substantially improves the estimated diffusion dynamics.

4. Comparative Analysis with Traditional Methods: Although the authors highlight the advantages of structure-aware tracking and ambiguity removal, the comparative analysis with traditional methods is insufficient. What are the performance differences between structure-aware methods and traditional methods under various conditions (e.g., different organelles, experimental setups)? The attempt to use FidlTrack to detect the binding of antibodies and the Amyloid Precursor Protein (APP) in the ER lumen is interesting but lacks enough analysis. Why is the displacement distribution a key factor for distinguishing binding? Why is the traditional method unable to detect this? It would be helpful to provide additional analysis or experiments to demonstrate the advantages of FidlTrack over traditional methods.

We have now added analyses of the performance of the FidlTrack approach comparing it with traditional methodologies in two new study cases (**new Fig. 5 & 6, added text page 6 paragraphs 5 & page 7 paragraphs 2 & 3 and Methods sections 1.10 and 1.11 page 18, videos S12-14**). We also added the comparisons between applying or not FidlTrack for the nanobody binding detection case (**modified Fig. 7, new Fig. S7a-c and added text page 8 paragraph 1**).

The new cases were selected given their challenging nature and their ability to clearly demonstrate how the enhancement afforded by the FidlTrack approach reveals the key biological events otherwise undetectable. In the nanobody binding showcase, the traditional method was

unable to reliably detect binding because of the frequent incorrect trajectories, erroneously linking bound (slow) and unbound (fast) molecules.

In more detail, we have added:

- **New panels Fig. S1c-e (added text page 4 paragraph 1 and Methods section 1.1.7 page 13)** show comparisons of fidelity improvement when FidlTrack uses different common tracking algorithms: nearest-neighbour, furthest-neighbour or u-track.
- **New panel Fig. S3d, e (added text page 4 paragraph 4)** shows a comparison of the recovered trajectory lengths and estimated diffusion coefficients before and after ambiguity removal in freespace simulations.
- **New panels Fig. 3c-e (added text page 5 paragraph 5)** show new simulation results for a parallel strips geometry, where structure-aware tracking improves fidelity up to 50% and reduces ambiguity by up to 78%.

On the question of the key factor for binding detection, the transition from membrane bound to a free soluble state results in a notable change in mobility, reflected in the shifted displacement distribution (see **updated scheme on Fig. 7b**).